# Decreasing trends of ammonia emissions over Europe seen from remote sensing and inverse modelling

**Ondřej Tichý[1], Sabine Eckhardt[2], Yves Balkanski[3], Didier Hauglustaine[3], Nikolaos Evangeliou[2],\***

[1] The Czech Academy of Sciences, Institute of Information Theory and Automation, Prague, Czech Republic.

[2] The Climate and Environmental Research Institute NILU, Department of Atmospheric and Climate Research (ATMOS), Kjeller, Norway.

[3] Laboratoire des Sciences du Climat et de l'Environnement (LSCE), CEA-CNRS-UVSQ, 91191, Gif-sur-Yvette, France.

**\*** Corresponding author: N. Evangeliou (Nikolaos.Evangeliou@nilu.no)

## Abstract

Ammonia (NH3), a significant precursor of particulate matter, not only affects biodiversity, ecosystems, soil acidification, but also climate and human health. In addition, its concentrations are constantly rising due to increasing feeding needs and the large use of fertilization and animal farming. Despite the significance of ammonia, its emissions are associated with large uncertainties, while its atmospheric abundance is difficult to measure. Nowadays, satellite products can effectively measure ammonia with low uncertainty and a global coverage. Here, we use satellite observations of column ammonia in combination with an inversion algorithm to derive ammonia emissions with a high resolution over Europe for the period 2013–2020. Ammonia emissions peak in Northern Europe, due to agricultural application and livestock management, in Western Europe (industrial activity) and over Spain (pig farming). Emissions have decreased by -26% since 2013 (from 5431 Gg in 2013 to 3994 Gg in 2020) showing that the abatement strategies adopted by the European Union have been very efficient. The slight increase (+4.4%) in 2015 is also reproduced here and is attributed to some European countries exceeding annual emission targets. Ammonia emissions are low in winter (286 Gg) and peak in summer (563 Gg) and are dominated by the temperature dependent volatilization of ammonia from the soil. The largest emission decreases were observed in Central and Eastern Europe (-38%) and in Western Europe (-37%), while smaller decreases were recorded in Northern (-17%) and Southern Europe (-7.6%). When complemented against ground observations, modelled concentrations using the posterior emissions showed improved statistics, also following the observed seasonal trends. The posterior emissions presented here also agree well with respective estimates reported in the literature and inferred from bottom-up and top-down methodologies. These results indicate that satellite measurements combined with inverse algorithms constitute a robust tool for emission estimates and can infer the evolution of ammonia emissions over large timescales.

# 1 Introduction

Ammonia ($NH_3$), the only alkaline gas in the atmosphere, constitutes one of the most reactive nitrogen species. It is produced from decomposition of urea, which is a rapid process when catalyzed by enzymes (Sigurdarson et al., 2018). The main sectors contributing to its production are livestock management and wild animals (Behera et al., 2013), biomass burning and domestic coal combustion (Fowler et al., 2004; Sutton et al., 2008), volcanic eruptions (Sutton et al., 2008), and agriculture (Erisman et al., 2007). Emissions from agricultural activity and livestock management represent over 80% of the total emissions (Crippa et al., 2020), while their regional contribution can reach 94% (Van Damme et al., 2018).

Once emitted, it is transported over short distances and deposited to water bodies, soil or vegetation with a typical atmospheric lifetime of a few hours (Evangeliou et al., 2021). It can then lead to eutrophication of water bodies (Stevens et al., 2010), modulate soil pH (Galloway et al., 2003) and «burn» vegetation by pulling water from the leaves (Krupa, 2003). It also reacts with the abundant atmospheric sulfuric and nitric acids (Malm, 2004) forming fine particulate matter (PM2.5) (Tsimpidi et al., 2007). While ammonia has a short atmospheric lifetime, PM2.5 resides significantly longer in the atmosphere, on the order of days to weeks (Seinfeld and Pandis, 2000), and hence is transported over longer distances. Accordingly, secondary PM2.5 can affect the Earth's radiative balance, both directly by scattering incoming radiation (Henze et al., 2012) and indirectly as cloud condensation nuclei (Abbatt et al., 2006). Its environmental effects include visibility problems and contribution to haze formation. Finally, PM2.5 affects human health, as it penetrates the human respiratory system and deposits in the lungs and alveolar regions (Pope and Dockery, 2006; Pope III et al., 2002) contributing to premature mortality (Lelieveld et al., 2015).

To combat secondary pollution, the European Union established a set of measures focusing on ammonia abatement, similar to the ones introduced by China (Giannakis et al., 2019). These measures aim at reducing ammonia emissions by 6% in 2020, relative to 2005. However, the lack of spatiotemporal measurements of ammonia over Europe makes any assessment of the efficiency of these measures difficult, as only bottom-up methods are used to calculate emission. These methods still show a slight increase (0.6% $y^{-1}$) up to 2018 mostly due to increasing agricultural activities (McDuffie et al., 2020). Such bottom-up approaches rely on uncertain land-use data and emission factors that are not always up to date, thus adding large errors to existing inventories.

During the last decade, satellite products have also become available to fill the gaps
created by spatially disconnected ground-based measurements. Data from satellite sounders
such as the Infrared Atmospheric Sounding Interferometer (IASI) (Van Damme et al., 2017),
the Atmospheric Infrared Sounder (AIRS) (Warner et al., 2017), the Cross-track Infrared
Sounder (CrIS) (Shephard and Cady-Pereira, 2015), the Tropospheric Emission Spectrometer
(TES) (Shephard et al., 2015), and Greenhouse Gases Observing Satellite (GOSAT) (Someya
et al., 2020) are publicly available. Most of them have been validated against ground-based
observations or complemented with other remote sensing products (Van Damme et al., 2015,
2018; Dammers et al., 2016, 2017, 2019; Kharol et al., 2018; Shephard et al., 2020; Whitburn
et al., 2016).
Accordingly, a few studies on ammonia emission calculations have been recently
published relying on 4D-Variational inversion schemes such as (Cao et al., 2022; Zhu et al.,
2013) or process based models (Beaudor et al., 2023; Vira et al., 2020). More recently, Sitwell
et al. (2022) proposed an inversion scheme for comparison between model profiles and satellite
retrievals using hybrid logarithmic and linear observation operator that attempts to choose the
best method according to the particular situation. In the present study, we use direct
comparisons between the CrIS ammonia retrievals and model profiles using the Least Squares
with Adaptive Prior Covariance (LS-APC) algorithm (Tichý et al., 2016), which reduces the
number of tuning parameters in the method significantly using variational Bayesian
approximation technique. We constrain ammonia emissions over Europe over the 2013–2020
period and validate the results against ground-based observations from EMEP (European
Monitoring and Evaluation Programme, https://emep.int/mscw/) (Torseth et al., 2012).

## 2 Methods

### 2.1 CrIS observations

To constrain ammonia emissions with inverse modelling, satellite measurements were
adopted from the Cross-Track Infrared Sounder (CrIS) onboard the NASA Suomi National
Polar-orbiting Partnership (S-NPP) satellite, which provides atmospheric soundings with a
spectral resolution of 0.625 cm$^{-1}$ (Shephard et al., 2015). CrIS presents improved vertical
sensitivity for ammonia closer to the surface due to the low spectral noise in the ammonia
spectral region (Zavyalov et al., 2013) and the early afternoon overpass that typically coincides
with high thermal contrast, which is optimal for thermal infrared sensitivity. The CrIS Fast
Physical Retrieval (CFPR) (Shephard and Cady-Pereira, 2015) retrieves ammonia profiles at

14 levels using a physics-based optimal estimation retrieval, which also provides the vertical sensitivity (averaging kernels) and an estimate of the retrieval errors (error covariance matrices) for each measurement. As peak sensitivity typically occurs in the boundary layer between 900 and 700 hPa (~ 1 to 3 km) (Shephard et al., 2020) and the surface and total column concentrations are both highly correlated with these boundary layer retrieved levels. The total column random measurement error is estimated in the 10–15% range, with total errors to be ~30% (Shephard et al., 2020). The individual profile retrieval levels show an estimated random measurement error of 10–30 %, with total random errors estimates increasing to 60 to 100% due to the limited vertical resolution (1 degree of freedom of signal for CrIS ammonia). These vertical sensitivity and error output parameters are also useful for using CrIS observations in applications (e.g. data fusion, data assimilation; model-based emission inversions; (Cao et al., 2020; Li et al., 2019)), as a satellite observational operator can be generated in a robust manner. The detection limit of CrIS measurements has been calculated down to 0.3–0.5 ppbv (Shephard et al., 2020). CrIS ammonia has been evaluated against other observations over North America with the Ammonia Monitoring Network (AMoN) (Kharol et al., 2018) and against ground-based Fourier transform infrared (FTIR) spectroscopic observations (Dammers et al., 2017) showing small bias and high correlations.

Daily CrIS ammonia (version 1.6.3) was put on a 0.5°×0.5° grid covering all of Europe (10°W–50°E, 25°N–75°N) for the period 2013–2020. Gridding was chosen due to the large number of observations (around 10,000 retrievals per day per vertical level), which made the calculation of source-receptor matrices (SRMs) computationally inefficient. Through gridding we limited the number of observation (and thus the number of SRMs to be calculated) to 2000 per day per vertical level. Sitwell et al. (2022) showed that the averaging kernels of CrIS ammonia are significant only for the lowest six levels (the upper eight have no influence onto the satellite observations) and therefore we considered only these six vertical levels (~1018-619 hPa). The gridding was performed by averaging the values that fall in each 0.5° resolution grid-cell daily over the 2013 – 2020 period of this study. This type of gridding was selected as previous experience with inverse distance weighting interpolation of satellite observations showed overestimated results of up to 100% (Evangeliou et al., 2021). In addition, the quality of gridding with respect to the averaging kernel of CrIS ammonia was evaluated by calculating the standard deviation of the averaged values (Supplementary Figure S 1). The latter shows that the kernel values within each grid-cell were very similar resulting in low gridded standard deviations, and thus low bias caused from the gridding (Supplementary Figure S 1).

## 2.2   A priori emissions of ammonia

We used as a priori emissions for ammonia in the inversion algorithm the ones calculated (i) from the most recent version of ECLIPSEv6 (Evaluating the CLimate and Air Quality ImPacts of Short-livEd Pollutants) (Klimont, 2022; Klimont et al., 2017) combined with biomass burning emissions from GFEDv4 (Global Fire Emission Dataset) (Giglio et al., 2013) hereafter "EC6G4", (ii) a more traditional dataset from ECLIPSEv5, GFEDv4 and GEIA (Global Emissions InitiAtive), hereafter "EGG" (Bouwman et al., 1997; Giglio et al., 2013; Klimont et al., 2017), (iii) emissions calculated from IASI (Infrared Atmospheric Sounding Interferometer) and a 1-dimensional box-model and a modelled lifetime (Evangeliou et al., 2021), denoted as "NE" and (iv) from the high resolution dataset of Van Damme et al. (2018) after applying a simple 1-dimensional box-model (Evangeliou et al., 2021), hereafter denoted as "VD". Given the large uncertainty in ammonia emissions illustrated in Figure 1, we calculated the average of these four priors (hereafter "avgEENV") to establish the a priori emissions used in this study.

## 2.3   Lagrangian particle dispersion model for the calculation of source-receptor matrices (SRMs) of ammonia

SRMs were calculated for each $0.5° \times 0.5°$ grid-cell over Europe (10°W–50°E, 25°N–75°N) using the Lagrangian particle dispersion model FLEXPART version 10.4 (Pisso et al., 2019) adapted to simulate ammonia. The adaptation of the code includes treatment for the loss processes of ammonia adopted from the Eulerian model LMDZ-OR-INCA (horizontal resolution of $2.5° \times 1.3°$ and 39 hybrid vertical levels) that includes all atmospheric processes and a state-of-the-art chemical scheme (Hauglustaine et al., 2004). The model accounts for large-scale advection of tracers (Hourdin and Armengaud, 1999), deep convection (Emanuel, 1991), while turbulent mixing in the planetary boundary layer (PBL) is based on a local second-order closure formalism. The model simulates atmospheric transport of natural and anthropogenic aerosols and accounts for emissions, transport (resolved and sub-grid scale), and dry and wet deposition. LMDZ-OR-INCA includes a simple chemical scheme for the ammonia cycle and nitrate particle formation, as well as a state-of-the-art $CH_4/NO_x/CO/NMHC/O_3$ tropospheric photochemistry (Hauglustaine et al., 2014). To calculate chemical loss of ammonia to PM2.5, after a month of spin-up, global atmospheric transport of ammonia was simulated for 2013–2020 by nudging the winds of the 6-hourly ERA Interim Reanalysis data (Dee et al., 2011) with a relaxation time of 10 days (Hourdin et al., 2006). Using the EGG inventory, we calculated the e-folding lifetime of ammonia in the model, which was adopted in

FLEXPART. We refer the reader to (Tichý et al., 2022) for a detailed description of the
formalism. Atmospheric linearities of the system and a full validation against ground-based
observation are also presented in the same paper.
FLEXPART releases computational particles that are tracked backward in time using
ERA5 (Hersbach et al., 2020) assimilated meteorological analyses from the European Centre
for Medium-Range Weather Forecasts (ECMWF) with 137 vertical layers, a horizontal
resolution of 0.5°×0.5° and one hour temporal resolution. FLEXPART simulates turbulence
(Cassiani et al., 2014), unresolved mesoscale motions (Stohl et al., 2005) and includes a deep
convection scheme (Forster et al., 2007). SRMs were calculated for 7 days backward in time,
at temporal intervals that matched satellite measurements and at spatial resolution of
0.25°×0.25°. This 7-day backward tracking is sufficiently long to include almost all ammonia
sources that contribute to surface concentrations at the receptors given a typical atmospheric
lifetime of about half a day (Van Damme et al., 2018; Evangeliou et al., 2021).

## 2.4   Inverse modeling algorithm

The inversion method used in the present study relies on optimization of the difference
between the CrIS satellite vertical profile observations, denoted as $v^{sat}$, and retrieved vertical
profile, $v^{ret}$. The latter are obtained by applying an instrument operator applied in logarithm
space (Rodgers, 2000) as follows:

$$\ln(v^{ret}) = \ln(v^a) + A(\ln(v^{true}) - \ln(v^a)) \qquad (1)$$

where $v^{ret}$ is the retrieved profile concentration vector, $v^a$ is a priori profile concentration
vector used in the satellite retrievals, $v^{true}$ is the hypothetical true profile concentration vector
supplied by the model ($v^{true} = v^{mod}$), and $A$ is the averaging kernel matrix (for each
0.5°×0.5° resolution grid-cell). Eq. (1) provides a useful basis for the calculation of the CrIS
retrievals if the retrieval algorithm is performing as designed, i.e., it is unbiased and the root
mean square error (RMSE) is within the expected variability. The $v^{mod}$ term can be written as:

$$v^{mod} = Mx \qquad (2)$$

for each grid-cell of the spatial domain, where $M$ is the grid-cell specific SRM calculated with
FLEXPART and $x$ is the unknown grid-cell specific emission vector. The SRM matrix $M$ is
calculated on circular surroundings around each grid-cell for computational efficiency. We
chose circles with a radius of approximately 445 km, equal to 4 degrees, which is shown to be
sufficient for reliable emission estimation and low sensitivity has been observed with this
choice. Since the vector $x$ is unknown, we replace it by a prior emission $x^a$ (see section 2.2) in
the initial step that is gradually refined iteratively based on the satellite observations.
The used inversion setup is based on iterative minimization of mismatch between $v^{sat}$
and $v^{ret}$ updating (iteratively) the emission $x$ such as below:
$$\arg \min_{x^a \to x} \left|\left| v^{sat} - v^{ret} \right|\right|_2^2 \qquad (3)$$

for each grid-cell of computational domain. The minimization problem is solved in two steps.
First, we construct the linear inverse problem for each year where $v^{ret}$ from the given
surroundings, denoted here as $S$, forms the block-diagonal matrix $v_S^{ret}$ while $v^{sat}$ from the
given surroundings form an associated observation vector $v_S^{sat}$. This forms the linear inverse
problem:
$$v^{sat} = v_S^{ret} q_S \qquad (4)$$

where the vector $q_S$ is a vector with coefficients denoting how $x^a$ needs to be refined to obtain
emission estimate vector $x$. All elements in Eq. 4 are affected by uncertainties originating from
both the observations and model, hence, we employ an inverse algorithm to solve Eq. 4 with
added regularization in the form of prior distributions with specific covariance models. For one
year, 6 vertical profiles, and 4 degrees radius, the size of the the block-diagonal matrix $v_S^{ret}$ is
13896 times 12, hence, the correction coefficient vector $q_S$ contain 12 values corresponding to
each month. We solve Eq. 4 using the least squares with adaptive prior covariance (LS-APC)
algorithm (Tichý et al., 2016). The algorithm is based on variational Bayesian methodology
assuming non-negative solution and favoring solution without abrupt changes and it minimizes
the use of manual tuning (Tichý et al., 2020). The method assumes the data model in the form
of:
$$p(v^{sat}) = N(v_S^{ret} q_S, R) \qquad (5)$$

where $N$ denotes the multivariate normal distribution and $R$ the covariance matrix assumed in
the form $R = \omega^{-1} I_p$, where $I_p$ is the identity matrix with ones on its diagonal and zeros
elsewhere, and $\omega$ is the unknown precision parameter on its diagonal. Following Bayesian
methodology, we assign a prior model to all unknown parameters, i.e. $\omega$ and $q_S$. Their prior
models are selected as:
$$p(\omega) = G(\vartheta_0, \rho_0) \qquad (6)$$

$$p(q_S) = tN(0, (LVL)^{-1}, [0, +\infty]) \qquad (7)$$

where $G(\vartheta_0, \rho_0)$ is the Gamma distribution (conjugate to the normal distribution) with prior
parameters $\vartheta_0, \rho_0$ selected to $10^{-10}$ achive non-informative prior. The second term follows
truncated normal distribution with positive support and with specific form of a precision matrix.
We assume the precision matrix in the form of modified Cholesky decomposition which allows
for tractability of estimation of its parameters, matrices $V$ and $L$. The matrix $V$ is diagonal with

unknown diagonal parameters and the matrix $L$ is lower bidiagonal with ones on the diagonal and unknown parameters on its sub-diagonal, formalized as vectors $v$ and $l$, respectively. These parameters are estimated within the method, while purpose of vector $v$ is to allow for abrupt changes in $q_S$, and vector $l$ to favor smooth estimates (see details in Tichý et al. (2016)). All model parameters ($\omega, q_S, v, l$) are estimated using the variational Bayes procedure where we obtain not only point-estimates, but their full posterior distributions.

Second, the grid-cell specific coefficient vector $q_S$ is propagated through Eq. 2 into Eq. 1 to refine a prior emission $x^a$ and obtain estimated unknown emissions $x$. To maintain stability of the method, we bound the ratio between prior and posterior emission elements to 0.01 and 100, respectively. This choice, motivated by Cao et al. (2020), omits unrealistically small or high emissions, however, the bounds are large enough to allow for new sources, as well as for attenuation of old sources. To introduce these boundaries is necessary since the problem in Eq. 1 is ill-conditioned and the propagation through the equation may lead to unrealistic values due to numerical instability. For this reason, these boundaries are needed and the sensitivity to the choice of the prior emission are studied in Section 3.3.

Note that CrIS data for some spatiotemporal elements are missing in the dataset. In these cases, we interpolated the missing data following the method proposed by D'Errico (2023), which solves a direct linear system of equations for missing elements, while the extrapolation behavior of the method is linear. Another strategy recently adopted in the literature has been to tackle the missing data using total variation methodology (see details in Fang et al., 2023); however, the method has been limited so far to its use on point-source release, hence we did not use it in this work.

## 3  Results

### 3.1   Emissions of ammonia in Europe (2013–2020)

We analyze the CrIS ammonia satellite observations for Europe (10°W–50°E, 25°N–75°N) over the 2013-2020 period on monthly basis to derive ammonia emissions using the inverse modelling methodology described in Section 2.4. The inversion algorithm is applied to each year of CrIS observations separately with the use of the avgEENV prior emission. Note that since a diurnal cycle is neither assumed in the Chemistry Transport Model, nor exists in the satellite observations from CrIS, daily emissions of ammonia do not represent a daily mean.

The overall resulting spatial distribution of the posterior emissions of ammonia (denoted as posterior_avgEENV) averaged for the whole period are displayed in Figure 2 (top-left). The

highest emissions occur in Northwestern Europe (including Northern Belgium, the Netherlands and northwestern Germany) and to a smaller extent in the Po Valley (Italy), and the Ebro Valley (Spain). Local maxima are also seen over Pulawy (Poland), South Romania and Kutina (Croatia) due to industrial applications (Clarisse et al., 2019; Van Damme et al., 2018). While ammonia emissions were not calculated high in the Po Valley (8 year average), it has been reported that in Lombardy, about 90% of the ammonia emissions there have been reported to originate from manure management (Lonati and Cernuschi, 2020). The Ebro Valley is characterized by intensive agricultural activities (Lassaletta et al., 2012; Lecina et al., 2010) and the Aragon and Catalonia regions by large pig farms (Van Damme et al., 2022). Finally, both Belgium and The Netherlands are countries in which intensive livestock activity is documented. It consists mostly of dairy cow, beef cattle, pig and chicken farming (Gilbert et al., 2018; Lesschen et al., 2011; Velthof et al., 2012).

Figure 2 (top-right) shows the annual posterior emissions discretized monthly for the whole period (solid line) compared to prior ammonia emissions (dashed line), averaged for the domain. Higher emissions than the prior ones were calculated, which is not necessarily attributed to emission increases over Europe, but rather to miscalculation of emissions in the prior bottom-up inventories that were used. A strong seasonal cycle is also observed peaking in the middle of each year (summer) of the study period, but for several of these years, the characteristic bimodal cycle also appears with another peak in spring (Beaudor et al., 2023).

To examine more closely the seasonal variability of ammonia emissions in Europe, we present the monthly posterior emissions of ammonia averaged for the whole study period (2013–2020) at the bottom-left panel of Figure 2 together with the prior ones. The total emissions for each month based on the map element size and length of the respective month were averaged for the whole study period. The same was done for each year in the bottom-right panel. The interannual variability over the period between 2013 and 2020, is also apparent in the monthly box and whisker plots of the posterior emissions. In addition, the spatial distribution of monthly ammonia emissions averaged for the eight-year period is given in Supplementary Figure S 2. It appears that ammonia emissions are very low in wintertime (DJF average: 286 Gg) over Europe and increase towards summer (JJA average: 563 Gg), due to temperature dependent volatilization of ammonia (Sutton et al., 2013), with the largest emissions occurring in August (601 Gg). Although a clear peak of fertilization in early spring is missing from the plot, emissions start to increase in early spring to peak in late-summer (Van Damme et al., 2022) corresponding to the start and end of the fertilization periods in Europe (Paulot et al., 2014). Fertilization is tightly regulated in Europe (Ge et al., 2020). It is only

allowed from February to mid-September in The Netherlands, while manure application is also only allowed during the same period depending on the type of manure and the type of land (Van Damme et al., 2022). In Belgium, nitrogen fertilizers are only allowed from mid-February to the end of August (Van Damme et al., 2022), so as in Germany (restricted in winter months) (Kuhn, 2017).

Finally, Figure 2 (bottom-right) shows the annual posterior emissions for the whole period with the annual total emissions for each year. We observe a significant decrease in ammonia posterior emissions over Europe during the 2013–2020 period. Emissions were estimated as 5431 Gg for 2013 decreasing to 4890 Gg in 2014. A minor increase can be seen in 2015 (5104 Gg), after which a significant decrease of 534 Gg (more than 10%) was estimated, followed by the nearly constant plateau at the levels between 4383 Gg in 2017, 4323 Gg in 2019 and finally to 3994 Gg in 2020. The gradual decrease in ammonia emissions over Europe since 2013 is also plotted spatially in Supplementary Figure S 3. It is evident that the restrictions and measures adopted by the European Union to reduce secondary PM formation were successful, as emissions in the hot-spot regions of Belgium, The Netherlands, Germany and Poland declined drastically over time. However, an increase of +4.4% was observed in 2015. It has been reported that ammonia emissions increased in 2015 and several European Union Member States, as well as the EU as a whole, exceeded their respective ammonia emission ceilings (EEA, 2017). The increase was reported to be +1.8% and was mainly caused by increased emissions in Germany, Spain, France, and the United Kingdom. This was caused by extensive use of inorganic nitrogen fertilizers (including urea application) in Germany, while increased emissions in Spain were driven by an increase in the consumption of synthetic nitrogen fertilizers and in the number of cattle and pigs (EEA, 2017). It should be mentioned that a false decrease of ammonia in 2020 due to the COVID-19 pandemic is calculated by the current methodology, mainly due to bias created by the decrease of $NO_x$ and $SO_2$ that are precursor species of the atmospheric acids, with which ammonia reacts (see Tichý et al., 2022).

## 3.2 Country-by-country ammonia emissions

Posterior annual emissions of ammonia for 2013–2020 are plotted for four European regions (Western, Central and Eastern, Northern and Southern Europe), accompanied by relative trends calculated as difference between year 2013 and 2020 divided by the average for the whole period, in the left panel of Figure 3, while the estimated seasonal variation of each region is shown on the right panels averaged over the whole eight-year period. Western Europe includes Ireland, Austria, France, Germany, Belgium, Andorra, Luxembourg, The Netherlands,

Switzerland, and United Kingdom; Central and Eastern Europe include Albania, Bosnia and Herzegovina, Bulgaria, Czechia, Croatia, Hungary, Belarus, Slovakia, North Macedonia, Montenegro, Poland, Romania, Moldova, Slovenia, Ukraine, and Serbia; Northern Europe is defined by Denmark, Estonia, Finland, Latvia, Lithuania, Faroe Islands, Norway, and Sweden; finally, Southern Europe includes Cyprus, Greece, Italy, Portugal, Spain.

The most significant decreases in ammonia emissions were estimated to be -38% in Central and Eastern Europe and -37% in Western Europe, respectively. Quantitatively, Central and Eastern Europe emissions were estimated to gradually drop from 2190 Gg in 2013 and to 1495 Gg in 2020 with a small increase in 2015 (2171 Gg) mainly because Germany, France and the United Kingdom missed their emission targets (EEA, 2017). Western European emissions of ammonia also declined constantly over time from 2041 Gg in 2013 to 1421 Gg in 2020. Smaller, yet significant, decreases were calculated over Northern Europe from 398 Gg in 2013 to 333 Gg in 2020 (-17%). Finally, Southern Europe exhibited a minor drop between years 2013 and 2014 (from 803 Gg in 2013 to 729 Gg in 2014) followed by a small increase until 2019 (from 729 to 803 Gg), and then decreased again in 2020 to 743 Gg. Overall, Southern European emissions decreased by -7.62%.

The seasonal cycle of ammonia was again characterized by the restrictions applied to the agricultural-related activities by the European Union member states (Figure 3, right panels). As such, emissions in Western, Central and Eastern and Southern Europe were very low in winter and started increasing when fertilization was allowed in early spring, whereas the increasing temperature towards summer increased volatilization and, thus, emissions of ammonia (Van Damme et al., 2022; Ge et al., 2020). Although much less marked than in other European regions due to lower prevailing temperatures and weaker agricultural applications, emissions in Northern Europe show the spring-summer temperature dependence. However, emissions were estimated to be double in winter rather following the cycle of $SO_2$ (Tang et al., 2020). Emission may increase in Northern Europe in winter because OH and $O_3$ concentrations are much lower, and the rate of converting $SO_2$ to sulfate much slower. This means that less sulfate is produced and thus more $NH_3$ stays in the gas form. Supplementary Figure S 4 shows prior emissions in Western, Central and Eastern, Northern and Southern Europe for EC6G4 and NE emission inventories. Both show the aforementioned increase in emissions during winter in Northeastern Europe. Specifically, the NE emissions that dominate the a priori emissions (avgEENV) as the highest inventory show an extreme winter peak in the north (emissions decline from 105 to 13 Gg). Therefore, there is a very strong dependence of the posterior

seasonality of ammonia in Northern Europe, which may be also influenced by the used prior
emissions, see uncertainty analysis in Section 3.3.

Country specific emissions of posterior ammonia on a monthly basis (eight-year average

emissions) are shown for 20 countries in Supplementary Figure S 5. For countries such as
Portugal, Spain, Italy, United Kingdom, The Netherlands, Belgium, Poland, Hungary,
Denmark, Belarus and Romania two peaks can be clearly seen in late spring and end of summer.
As discussed before, these peaks coincide with the two main fertilization periods in Europe
(Paulot et al., 2014). However, it is expected that ammonia abundance is high throughout the
entire spring–summer period (e.g., Greece, France, Germany, Czechia, Ukraine and Bulgaria)
due to agricultural activity and temperature dependent volatilization (Sutton et al., 2013).
Ammonia emissions in Finland, Sweden and Norway are smaller than in the rest of Europe and
show a reverse seasonality.

## 3.3   Uncertainties in ammonia's posterior emissions

For the calculation of uncertainty of the estimated posterior emissions two different

approaches were used. The first approach is based on uncertainty arising as a result of the
inversion methodology. The standard deviation is calculated from posterior estimate which is
in the form of Gaussian distribution such as
$$p_{\mathrm{posterior}}(x_i) = N(\mu_i, \sigma_i^2) \qquad (8),$$

where $N$ denotes normal (Gaussian) distribution and posterior parameters $\mu_i$ and $\sigma_i$ are results
of inversion for each element of the spatiotemporal domain. The uncertainty associated with
any given spatial element is then a property of Gaussian distribution defined with the square
root of summed squared standard deviations:
$$\sigma_{\mathrm{location}} = \sqrt{\sum_t \sigma_{\mathrm{location},t}^2} \qquad (9)$$

Here, $\sigma_{\mathrm{location},t}^2$ denotes the estimated variance of the emissions for given coordinates and time
period; we consider uncertainty calculated as 2σ standard deviations, i.e. 95% of the values lay
inside the interval with the center in the reported emissions surrounded by the reported
uncertainty.

The second approach is based on ensemble of the used prior emissions as an input for the

inversion. The different ensemble members are built from five prior emissions (see Figure 1)
while the uncertainty is calculated as the standard deviation of five resulting posterior
emissions.
The calculated posterior uncertainty for our spatial domain and studied period (2013–
2020) is shown in Figure 4 for Gaussian posterior (left) and for ensemble of prior emissions
(right). The uncertainty associated with Gaussian posterior for each year of the study period are
depicted in Supplementary Figure S 6. The absolute uncertainty of Gaussian posterior ammonia
emissions reaches a maximum of 23.3 ng m$^{-2}$ s$^{-1}$ or about 39% (relative value, calculated based
on related maximum of posterior emissions). The uncertainty based on prior ensemble reaches
a maximum of 60.2 ng m$^{-2}$ s$^{-1}$ which is equal to about 101% based on related maximum of
posterior emissions. In general, the pattern of both posterior uncertainties, Gaussian posterior
and prior ensemble respectively, are in agreement in theirs patterns and follow the one of the
posterior emissions, with the highest values over (i) Belgium, the Netherlands, and Germany
due to livestock, farming, and agricultural activity; (ii) Poland, South Romania and Croatia due
to industrial applications; (iii) Catalonia due to pig farming; (iv) West France due to manure
application. Nevertheless, the obtained posterior uncertainty remains low, and this depicts the
robustness of the methodology used and the calculated posterior emissions of ammonia.
**3.4   Validation of posterior emissions**
As shown in Eq. 3 (Section 2.4), the inversion algorithm minimizes the difference
between the satellite observations ($v^{sat}$) and the retrieved ammonia concentrations ($v^{ret}$). The
latter is a function of different satellite parameters (e.g., averaging kernel sensitivities) and
modelled ammonia concentrations using a prior dataset ($v^{mod}$ or $v^{true}$) as seen in Eq. 1. The
overall result is always propagated to $v^{mod}$ iteratively, each time updating the prior emissions
to obtain posterior ammonia. As specified in CrIS guidelines, modelled concentrations ($v^{mod}$)
cannot be directly compared with satellite data ($v^{sat}$), while comparing $v^{sat}$ with $v^{ret}$ is not a
proper validation method, because the comparison is performed for satellite observations that
were included in the inversion (dependent observations), and the inversion algorithm has been
designed to reduce the $v^{sat}$-$v^{ret}$ mismatches. This means that the reduction of the posterior
retrieved concentration ($v^{ret}$) mismatches to the observations ($v^{sat}$) is determined by the
weighting that is given to the observations with respect to $v^{ret}$. A proper validation of the
posterior emissions is performed against observations that were not included in the inversion
(independent observations).
For these reasons, we compare modelled posterior concentrations of ammonia ($v^{mod}$) at
the surface with ground-based observations over Europe from the EMEP (European Monitoring
and Evaluation Programme, https://emep.int/mscw/) network (Torseth et al., 2012). The
measurements are open in public and can be retrieved from https://ebas.nilu.no. We used
measurements for all years between 2013 and 2020 from an average of 53 stations with 2928
observations for each station covering all Europe (Supplementary Figure S 7). The comparison
is plotted for each of the 53 stations separately on a Taylor diagram in Figure 5. For all stations,
the Pearson's correlation coefficient increased for the posterior ammonia (coloured circles)
increased as compared to the prior one (coloured squares) reaching above 0.6 at several stations,
while the normalized root mean square error (nRMSE) and standard deviation were kept below
2 (unitless) and 2 μg m-3, respectively, in almost all stations (except SI0008 in Slovenia).

To further show how posterior emissions of ammonia affect modelled concentrations, we

chose six stations (DE0002 in Germany, NO0056 in South Norway, ES0009 in Spain, NL0091
in the Netherlands, HU0002 in Hungary and PL0005 in Poland) from the EMEP network
(highlighted in red in Supplementary Figure S 7), and we plot prior and posterior concentrations
against ground-based ammonia over time for the whole study period (2013–2020) in
Supplementary Figure S 8. Given the long period of plotting, we average observations every
week and modelled concentrations every month for a more visible representation of the
comparison. To evaluate the comparison, we calculate a number of statistic measures, namely
nRMSE, the normalized mean absolute error (nMAE) and the root mean squared logarithmic
error (RMSLE) as defined below:
$$nRMSE = \frac{\sqrt{\sum_{i=1}^{n} \frac{1}{n}(m_i - o_i)^2}}{\frac{1}{n}\sum_{i=1}^{n} o_i} \qquad nMAE = \frac{\sum_{i=1}^{n}|m_i - o_i|}{\sum_{i=1}^{n} o_i}$$
$$RMSLE = \sqrt{\frac{1}{n}\sum_{i=1}^{n}(\log m_i - \log o_i)^2} \qquad (10)$$
where $n$ is sample size, $m$ and $o$ the individual sample points for model concentrations and
observations of ammonia indexed with $i$. As one can see in Supplementary Figure S 8, all
statistics were improved in all six stations and posterior concentrations were closer to the
observations. However, individual peaks were in many cases misrepresented in the model.
Whether this is a result of the measurement technique or the fact that local sources cannot be
resolved at the spatiotemporal resolution of CTM and FLEXPART (given the short lifetime of
atmospheric ammonia) needs further research. The best results were obtained at station ES0009
(Spain), where model captures the seasonal variation of the observations during the whole study
period (2013–2020). In all other stations, the seasonality is maintained albeit steep peaks in the
observations are lost.

As explained in section 1, ammonia reacts with the available atmospheric acids producing

secondary aerosols (Seinfeld and Pandis, 2000). Therefore, its presence and lifetime in the
atmosphere is driven by the atmospheric acids and their precursors, SO2 and NO2. Changes in

atmospheric levels of these substances have a significant impact on the lifetime of ammonia and its emissions, as highlighted in Tichý et al., (2022). Therefore, it is clear that a wrong representation of trends in modelled SO2 and NO2 will lead to systematic biases in the estimated ammonia emission trends. To further demonstrate that the modelling system correctly represents the trends in SO2 and NO2, we compare ground-based observations of these two species from the EMEP network (https://emep.int/mscw/) against modelled concentrations.

The comparison is shown in Supplementary Figure S 9 for six random EMEP stations for different years for each of NO2 and SO2. The full comparison of the two datasets of observations is plotted in scatterplots of modelled versus measured surface concentrations for NO2 and SO2 for all the study period (2013-2020) in Supplementary Figure S 10. A total number of 3,368,660 for SO2 and 4,252,592 for NO2 was used in the validation. It is evident that the seasonal variation of the modelled surface concentrations and their magnitude are both represented very well in the model for NO2 and SO2. nRMSE was 0.12 – 0.19 for NO2 and 0.09 – 0.25 for SO2, nMAE 0.39 – 0.94 for NO2 and 0.48 – 1.2 for SO2, RMSLE was 0.25 – 0.49 for NO2 and 0.11 – 0.33 for SO2 in the six stations (Supplementary Figure S 9). For over 4.2 and 3.3 million measurements that were used in this validation of NO2 and SO2 concentrations for 2013 – 2020 study period, nRMSE values were 0.05 and 0.02, nMAE 0.74 and 1.0 and RMSLE 0.50 and 0.40 for NO2 and SO2, respectively (Supplementary Figure S 10).

# 4 Discussion

## 4.1 Comparison with emissions inferred from satellite observations

We compared our posterior estimates with two recently published studies on ammonia emission in Europe (Cao et al., 2022; Luo et al., 2022). Luo et al. (2022) used IASI observations for the period 2008 to 2018 to estimate ammonia emissions in a global domain. Their method was based on updating prior emissions with correction terms computed using differences between observed and simulated ammonia columns combined with calculated ammonia lifetimes. The key indicators calculated for the European domain in Luo et al. (2022) are a linear trend for the 2008–2018 period, average annual emissions, and relative trends. Note that we compare our eight-year period with a decade in Luo et al. (2022). The comparison is depicted in Figure 6. Our estimates (Figure 6, left panel) are in good agreement with those calculated by Luo et al. (2022). The linear trend was estimated as -1.27 Tg for the period by Luo et al. (2022), while our estimate is -1.44 Tg. The spatial distribution of the trend is also

given in Figure 6 (left panel). The key decrease is observed mainly in France, Germany, and middle Europe, while the increasing trend is observed mostly in Spain, parts of Italy, and Greece. The average annual ammonia emission for the European domain in Luo et al. (2022) was estimated to be 5.05 Tg while our estimate is 4.63 Tg. Our lower estimate (by approximately 8%) might be attributed to use of a more recent period considered in our study, but both methods agree that the trend in Europe is negative. The relative decrease estimated by Luo et al. (2022) is -25.1%, while we calculate -31.02%, which is again in very good agreement.

Cao et al. (2022) used CrIS observations for the year 2016 in order to estimate ammonia emissions for 25 European Union members (EU25), namely Austria, Belgium, Bulgaria, Croatia, Republic of Cyprus, Czech Republic, Denmark, Estonia, France, Germany, Greece, Hungary, Ireland, Italy, Latvia, Lithuania, Luxembourg, Malta, Netherlands, Poland, Portugal, Romania, Slovakia, Slovenia, and Spain. The method was tested with uni-directional and bi-directional flux schemes. The uni-directional dry deposition scheme assumes only air to surface exchange of ammonia ignoring changes in environmental conditions, while the bi-directional scheme captures dynamics in measured ammonia fluxes. Total estimated ammonia emissions for the EU25 region by the uni-directional scheme (posterior_uni) and the bi-directional scheme (posterior_bi) were reported as 3534 Gg N y−1 and 2850 Gg N y−1, respectively. The posterior_bi estimate is very close to our estimate for EU25 for the year 2016, which is 2712 Gg N y−1, while the posterior_uni is approximately 30% higher. A uni-directional dry deposition scheme ignores the impacts of changes in environmental conditions (e.g., soil temperature, soil wetness, soil pH, fertilized condition, and vegetation type) on ammonia emissions from fertilized soil and crops (volatilization), which likely lead to high biases in top-down estimates. Ammonia in LMDz-OR-INCA model, that was used to capture ammonia's losses, resembles a partially bi-directional treatment, where emissions and deposition are both possible at the same time without any use of a compensation point; this may explain this 30% difference.

The detailed EU25 emissions for the year 2016 are displayed in Figure 6 (right panel) for posterior_uni (red), posterior_bi (yellow), our post_avgEENV (blue), and priors used by Cao et al. (2022) and in our study (dashed red and blue, respectively). As seen from Figure 6, our posterior estimates (post_avgEENV) have more similar characteristics with posterior_bi, with monthly difference to be less than factor of 2 positive or negative from Cao et al. (2022). Note that the posterior_uni estimates are always a factor of 3 higher than our posterior estimates for ammonia emissions. The main differences can be observed during February-March and October-November periods where our estimates are generally lower than those from Cao et al.

(2022). Finally, Cao et al. (2022) reported more of a springtime peak likely associated with crop fertilization, whereas in this study the peak is towards the warmer season (volatilization) due to livestock sources.

**4.2 Assessment of ammonia's atmospheric linearities**

Ammonia is a particularly interesting substance due to its affinity to react with atmospheric acids producing secondary aerosols. In most cases, it is depleted by sulfuric and nitric acids. However, when relative humidity is high and particles are aqueous, sulfate reacts with ammonia and decreases, while the equilibrium vapor pressure of ammonia with nitric acid increases shifting the reaction towards production of free ammonia (Seinfeld and Pandis, 2000). The former reaction is a rare event and lots of prerequisites must be fulfilled to take place.

Supplementary Figure S 11a shows the frequency distribution of gain (production of free ammonia - negative numbers) or loss (production of sulphate/nitrate – positive numbers) due to all chemical processes in the inversion domain (10°W–50°E, 25°N–75°N), for the study period (2013 – 2020) and the lowest six sigma-p vertical levels (~1018-619 hPa,  see averaging kernels in section 2.1) (Sitwell and Shephard, 2021). The figure shows mostly positive numbers indicating that atmospheric ammonia reacts towards secondary aerosol formation. The spatial distribution of gain/loss of ammonia is shown in Supplementary Figure S 11b. The pixels indicating production of gaseous ammonia are located in marine regions, where we chose to not perform inversions, as they are an order of magnitude lower (Bouwman et al., 1997), thus less significant. No continental pixels showing gain of ammonia were detected, which would cause simulated backwards in time to fail with our Lagrangian model (see next paragraph). Our approximation, although simplistic, provides computational efficiency when simulating SRMs in backward mode using FLEXPART (Pisso et al., 2019).

Seibert and Frank (2004) reported that standard Lagrangian particle dispersion models cannot simulate non-linear chemical reactions. First-order chemical reactions, where the reaction rates can be prescribed, are also linear. Non-linear chemistry cannot be calculated because neither the background chemistry is modeled nor is the coupling of the tracked plume (forward or backward) to this background. Technically, the SRM in FLEXPART is calculated for a receptor with a certain mean mixing ratio ($\chi$) and an emitting source ($q_{i,n}$) in a certain discretization of the space (index $i$) and time (index $n$), as:

$$\frac{\chi}{q_{i,n}} = \frac{1}{J}\sum_{j=1}^{J} \Delta t_{i,j,n} \frac{p_{j,n}}{\rho_{i,n}} \qquad (11)$$

where $J$ is the total number of backward trajectories (particles index $j$ ) originating from the position of the receptor $\chi$ and ending at a certain discretized time (index $n$) in certain discretized space position (index $i$) for a time interval $\Delta t_{i,j,n}$ , and where the air density is $\rho_{i,n}$. The further function $p_{j,n}$ ($p_{j,n} \leq 1$) represents the relative (to the initial receptor state) decay of the mass value in the particle in its travel from the receptor to the discretized space time interval $(j, n)$ due to any linear decay process (e.g. deposition, linear chemical decay) for a perfectly conserved scalar $p_{j,n} = 1$. So, for linear decaying species a direct SRM can be calculated explicitly among all relevant receptor points and all positions in space and time. The existence of the SRM (***H***), linking directly mixing ratios at the receptor points with emissions, is the prerequisite to apply simple inversion algorithms, such as the one in the present study.

Inversion of observations to obtain emissions for non-linear chemically reactive species entails the need of a chemistry transport model (CTM) forward (and its adjoint backward) in time from time $t_0$ to time $t_{\square}$ evolving the full state of the atmosphere, in relation to the emissions and boundary conditions. Subsequently, a cost function is evaluated by an iterative descent gradient method that implies running the adjoint of the forward model (Fortems-Cheiney et al., 2021). Note that an iterative algorithm means that the forward and adjoint models run several times in sequence until the estimated minimum of the cost function is reached.

To overcome these complexities, we examine the linearities of our method and show that FLEXPART simulates ammonia efficiently, we evaluate modelled ammonia against ground-based measurements of ammonia from EMEP (https://emep.int/mscw/) in Europe, EANET (East Asia acid deposition NETwork) in Southeastern Asia (https://www.eanet.asia/) and AMoN (Ammonia Monitoring Network in the US, AMoN-US; National Air Pollution Surveillance Program (NAPS) sites in Canada) in North America (http://nadp.slh.wisc.edu/data/AMoN/). The SRMs for ammonia express the emission sensitivity (in seconds) and yield modelled concentrations at the receptor point when coupled with gridded emissions from EGG (in kg m-2 s-1, see section 2.2) at the lowest model level (100 m). To check the consistency of the proxy used in the SRMs of ammonia, we also simulated surface concentrations of ammonia with FLEXPART in forward mode using the same emissions (EGG). We have chosen two random ground-based stations from each of the three measuring networks (EMEP, EANET, AMoN) to compare modelled concentrations. For consistency, we also plot the resulting surface concentrations from the LMDz-OR-INCA model (Supplementary Figure S 12).

Modelled concentrations (forward and backward FLEXPART and the CTM LMDz-OR-INCA) at each station have been averaged to the temporal resolution of the observations. Supplementary Figure S 13 shows Taylor diagrams of the comparison between FLEXPART simulated concentration in forward and backward mode. Plotting backward versus forward results is a common procedure to infer whether a Lagrangian model produces reasonable results (Eckhardt et al., 2017; Pisso et al., 2019). In general, the forward and backward simulations show very good agreement for the depicted receptor points. For example, ammonia concentration at stations AL99, CA83, and VNA001 (Supplementary Figure S 12) are simulated similarly, and the mean concentrations are almost identical in the forward and backward modes. However, during some episodes there can be notable differences (e.g., at DE0002R) as seen before (Eckhardt et al., 2017). The main reason is that the backward calculations always give more accurate results as the number of particles released at the receptor is much higher in backward mode than in forward mode; the particles are targeted to a very small location in backward, whereas in forward mode the particles are distributed equally on a global scale and therefore less particles represent each receptor location. Another reason is that transport and especially turbulent processes are parametrized by random motion, which are different for each FLEXPART simulation. Finally, the coordinate system for defining the height layer above ground depends on the meteorological field which is read at the start of the simulation, and this can also cause small deviations. The Taylor diagram for the respective comparison (Supplementary Figure S 13) show high Pearson's correlation coefficients (>0.7), low standard deviations (<1 µg N m-3) and root means square errors (RMSEs <0.7 µg N m-3).

## 4.3   Limitations of the present study

The latest Commission Third Clean Air Outlook published in December 2022 (EC, 2022), which is based on data reported by the EEA (https://www.eea.europa.eu/data-and-maps/dashboards/necd-directive-data-viewer-7), concluded (p. 2) that emissions of ammonia in recent years remain worryingly flat or may have increased for some member states. The assessment covers the period we investigated in the present manuscript (2013 – 2020) and shows (for the EU27) a reduction in ammonia emissions of only 2% that is far smaller than that we calculated here (26%). The consistency of our results with those calculated with similar methodologies (Cao et al., 2022; Luo et al., 2022) urges us to believe that such differences in ammonia trends are the result of differences between bottom-up and top-down methodologies.

628       Another reason for the difference in emissions between EC (2022) and this study might

be the fact that we used both nigh- and day-time CrIS observations. The night-time observations
use the exact same retrieval approach as the day-time ones. Any unknown bias would only be
present, if atmospheric conditions are such that it makes the retrievals more challenging during
night overpasses (e.g., thermal inversions causing an radiative emission layer at the surface,
etc.). Note that CrIS night-time observations have not been validated yet.

CrIS ammonia retrievals are performed in the logarithmic space and are expressed by Eq.

(1). When solving the cost function in the inverse modelling algorithm, we minimize the model-
observation mismatches, which in the present case is $v^{ret}$ (calculated using modelled
concentrations, $v^{true} = v^{mod}$, in Eq. (1) after applying prior emissions) with the CrIS
observations, $v^{sat}$. Then, iteratively, the algorithm updates $v^{mod}$ to calculate the posterior
emissions through Eq.(1). The logarithmic nature of Eq.(1) causes (i) the optimization of the
modelled concentrations ($v^{mod}$) to be tiny, and (ii) the trend of the posterior emissions to have
a very large dependency of the prior column used in CrIS ($v^a$). The latter is shown very clearly
in Figure 2 (lower right).

## 5  Conclusions

Today, a large debate takes place about ammonia abatement strategies for Europe, but

also for Southeastern Asia, in an effort to reduce secondary formation and, thus, mitigate
climate crisis (van Vuuren et al., 2015). These strategies include (a) low nitrogen feed by
reducing ammonia emissions at many stages of manure management, from excretion in
housing, through storage of manure to application on land, also having positive effects on
animal health and indoor climate (Montalvo et al., 2015); (b) low emission livestock housing,
which focuses on reducing the surface and time manure is exposed to air by adopting rules and
regulations regarding new livestock houses (Poteko et al., 2019); (c) air purification by
adopting technologies to clean exhaust air from livestock buildings (Cao et al., 2023) and
others. Here we used satellite observations from CrIS and a novel inverse modelling algorithm
to study the spatial variability and seasonality of ammonia emissions over Europe. We then
evaluated the overall impact of such strategies on the emissions of ammonia for the period
2013–2020. The main key messages can be summarized below:
• The highest emissions over the 2013–2020 study period occur in North Europe (Belgium,

the Netherlands and northwestern Germany). At a regional scale, peaks are seen in Western

Europe (Poland, South Romania and Croatia) due to industrial activities, in Spain (Ebro

Valley, Aragon, Catalonia) due to agricultural activities and farming, in Belgium and The
Netherlands due to livestock activity (dairy cow, beef cattle, pig and chicken farming).
• Ammonia emissions are low in winter (average: 286 Gg) and peak in summer (average:
563 Gg), due to temperature dependent volatilization of ammonia, while a notable peak
attributed to fertilization can be seen in early spring during some years.
• Over the 2013–2020 period, European emissions of ammonia decreased from 5431 Gg in
2013 to 3994 Gg in 2020 or about -26%. Hence, the restrictions adopted by the European
Union members were effective in reducing secondary PM formation.
• A slight emission increase of +4.4% in 2015 appears for several European Union Member
States (Germany, Spain, France, and the United Kingdom) who exceeded the respective
ammonia emission targets. Part of the 2020 ammonia decrease might be attributable to the
COVID-19 pandemic restrictions.
• The largest decreases in ammonia emissions were observed in Central and Eastern Europe
(-38%, 2190 Gg in 2013 to 1495 Gg in 2020) and in Western Europe (-37%, 2041 Gg in
2013 to 1421 Gg in 2020). Smaller decreases were calculated in Northern Europe (-17%,
398 Gg in 2013 to 333 Gg in 2020) and Southern Europe (-7.6%, from 803 Gg in 2013 to
to 743 Gg in 2020).
• The maximum calculated absolute uncertainty of Gaussian posterior model was 23.3 ng m$^-$
$^2$ s$^{-1}$, or about 39% (relative value) and calculated maximum based ensemble of prior
emissions was 60.2 ng m$^{-2}$ s$^{-1}$, or about 101% following the spatial distribution of the
posterior emissions.
• Comparison of the concentrations calculated with prior and posterior ammonia emissions
against independent (not used in the inversion algorithm) observations showed improved
correlation coefficients and low nRMSEs and standard deviations. Looking at timeseries
of six randomly selected stations in Europe, we also found that posterior surface
concentrations of ammonia were in accordance with the ground-based measurement, also
following the observed seasonal trends.
• Our results agree very well with those from Luo et al. (2022) (decreasing trend: -1.44
versus -1.27 Tg, annual European emissions: 4.63 versus 5.05 Tg) and those from Cao et
al. (2022) following their methodology (their posterior_bi estimate for EU25 and year 2016
was 2850 Gg N y−1, while we calculate 2712 Gg N y−1).
•   The relatively low posterior uncertainty and improved statistics in the validation of the
posterior surface concentrations denote the robustness of the posterior emissions of
ammonia calculated with satellite measurements and our adapted inverse framework.


*Data availability.* The data generated for the present paper can be downloaded from ZENODO
(https://doi.org/10.5281/zenodo.7646462). FLEXPARTv10.4 is open access and can be
downloaded from https://www.flexpart.eu/downloads, while use of ERA5 data is free of
charge, worldwide, non-exclusive, royalty-free and perpetual. The inversion algorithm LS-APC
is open access from https://www.utia.cas.cz/linear_inversion_methods. CrIS ammonia can be
obtained by request to Dr. M. Shephard (Mark.Shephard@ec.gc.ca). EMEP measurements are
open in https://ebas.nilu.no. FLEXPART SRMs for 2013–2020 can be obtained from the
corresponding author upon request.

*Competing interests.* The authors declare no competing interests.
*Acknowledgements.* This study was supported by the Research Council of Norway (project
ID: 275407, COMBAT – Quantification of Global Ammonia Sources constrained by a
Bayesian Inversion Technique). We kindly acknowledge Dr. M. Shephard for providing CrIS
ammonia. This work was granted access to the HPC resources of TGCC under the allocation
A0130102201 made by GENCI (Grand Equipement National de Calcul Intensif). O. Tichý
was supported by the Czech Science Foundation, grant no. GA20-27939S. Special thanks to
Dr. M. Cassiani for explaining Lagrangian modelling basics in section 4.2.


*Author contributions.* O.T. adapted the inversion algorithm, performed the calculations,
analyses and wrote the paper. S.E. adapted FLEXPARTv10.4 to model ammonia chemical loss.
Y.B. and D.H. set up the CTM model and performed the simulation, the output of which was
used as input in FLEXPART. N.E. performed the FLEXPART simulations, contributed to
analyses, wrote and coordinated the paper. All authors contributed to the final version of the
manuscript.

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

 **FIGURES & LEGENDS**

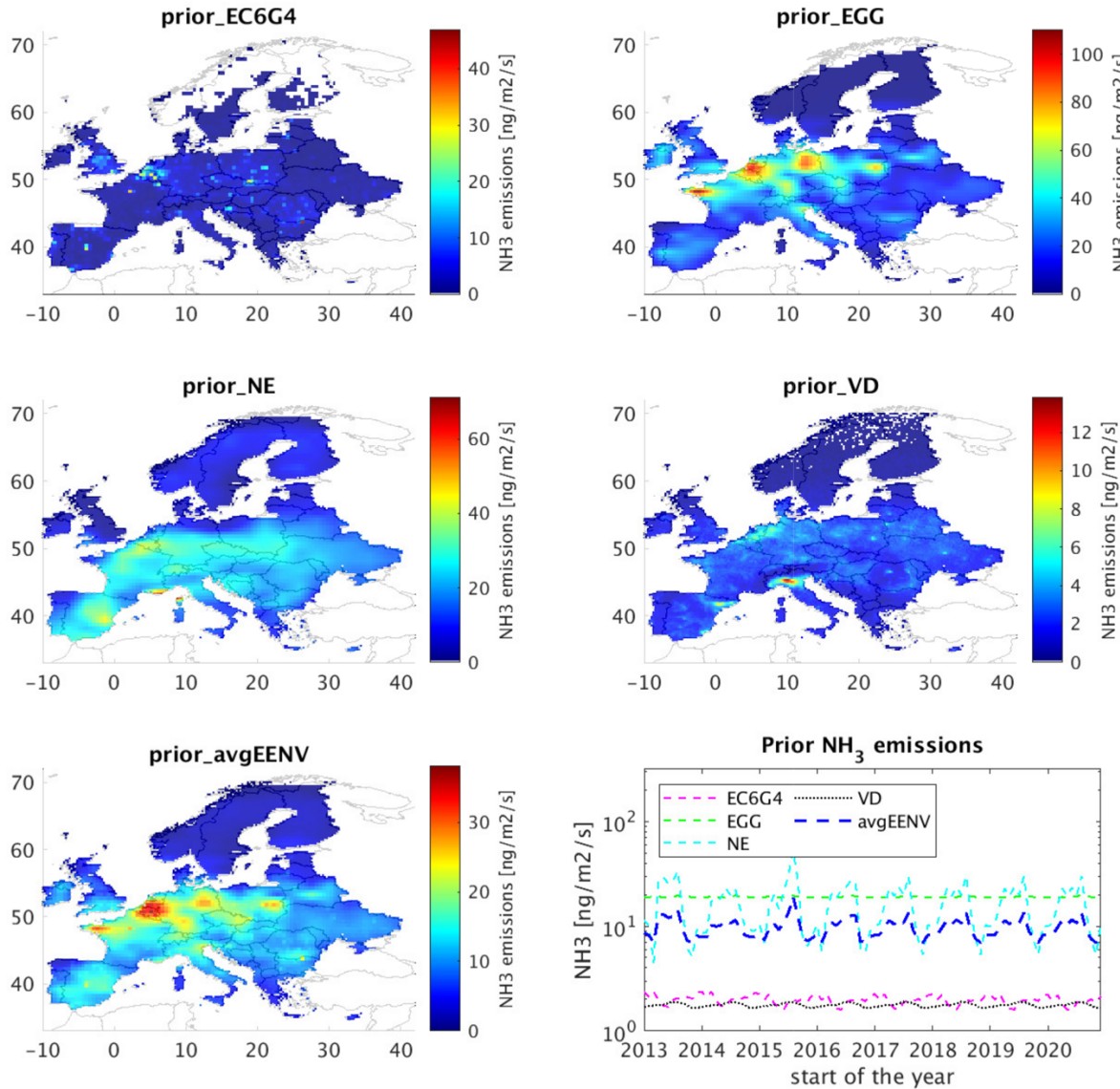


**Figure 1.** Four ammonia prior emissions (EC6G4, EGG, NE, VD) are displayed in the first two
rows. The combined prior (avgEENV) is displayed in the bottom left. The temporal variability
of all five prior emissions is given in bottom right.


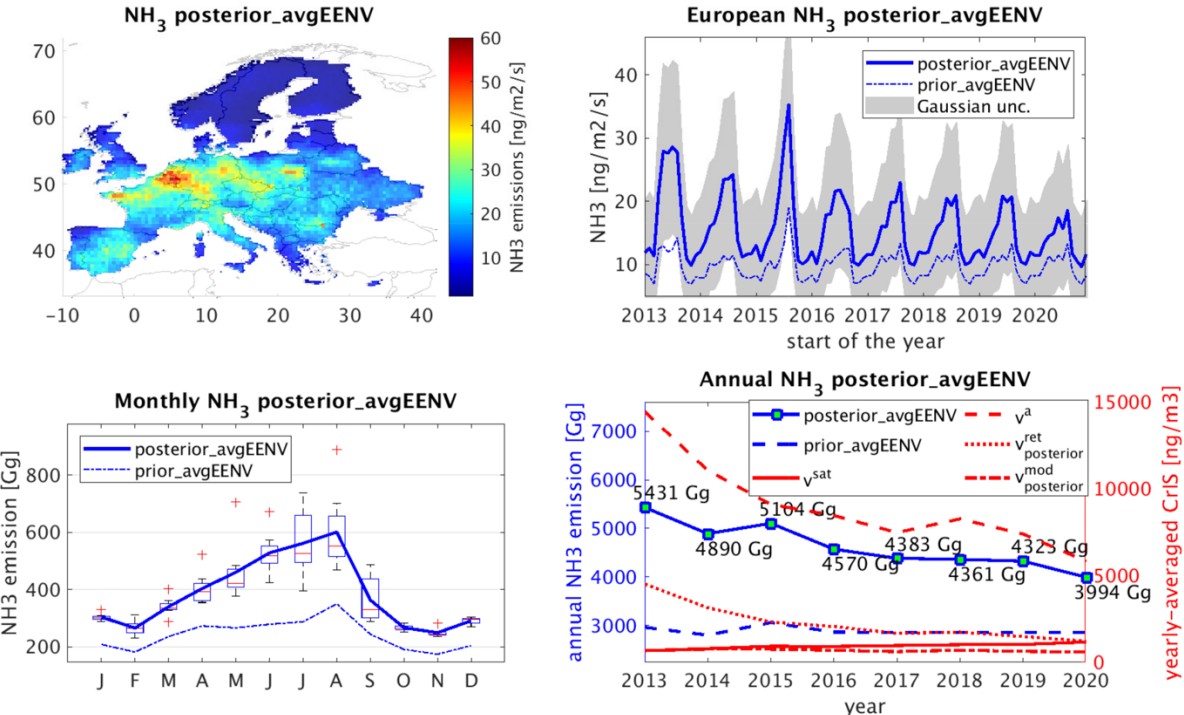


**Figure 2.** The spatial distribution of posterior ammonia emissions (posterior_EENV, top-left)
together with its temporal distribution (top-right). The Gaussian uncertainty of the posterior
emissions is also plotted. Monthly average ammonia emissions are shown in bottom-left graph.
The monthly average posterior emissions over the studied period are accompanied by the box
plot where the red line indicates the median, the bottom and top edges of the Boxes indicating
the 25[th] and 75[th] percentiles, respectively, and the whiskers extend to the most extreme data
points not considered as outliers, which are denoted using red crosses. Solid blue lines refer to
the posterior ammonia emissions, while dashed ones to the prior emissions (avgEENV).
Finally, annual average ammonia emissions are also plotted (bottom-right). Except for the
annual average emission dosages that are shown in blue, we also depict the elements that were
used to calculate $v^{ret}_{posterior}$, namely $v^a$ and $v^{mod}_{posterior}$ (see Eq. 1) that were compared with
$v^{sat}$.

1094

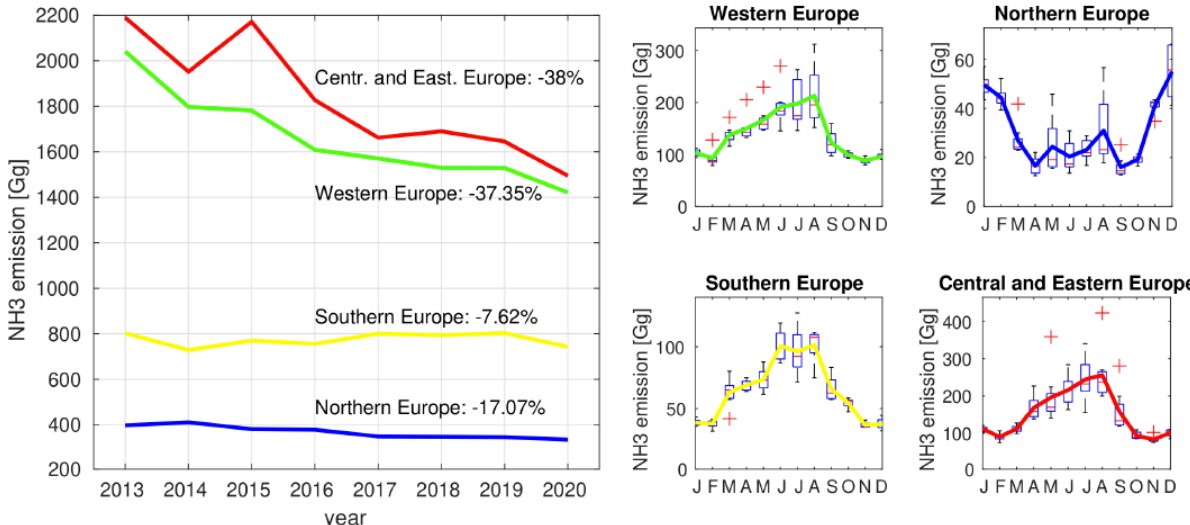

1095

**Figure 3.** Left: Annual posterior emissions of ammonia in Southern (yellow), Western (green), Northern (blue), and Central and Eastern (red) Europe. Right: Monthly average posterior emissions of ammonia accompanied by box plots, where the red line indicates the median, the bottom and top edges of the box indicate the 25$^{th}$ and 75$^{th}$ percentiles, respectively, and the whiskers extend to the most extreme data points (not considered outliers), which are represented using red crosses.

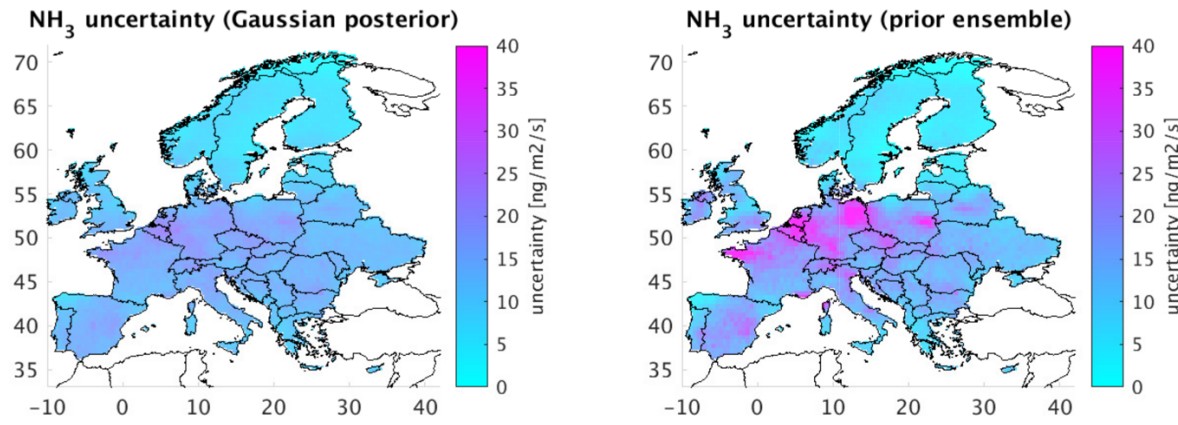

1103

**Figure 4.** Absolute uncertainty of posterior emissions of ammonia calculated as 2σ (left panel) and from a member ensemble (right panel) comprising posterior emissions calculated with five different priors (**Figure 1**) averaged for the whole study period 2013–2020.

1107

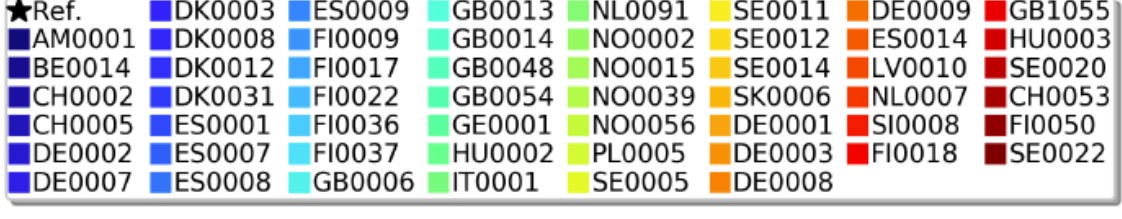

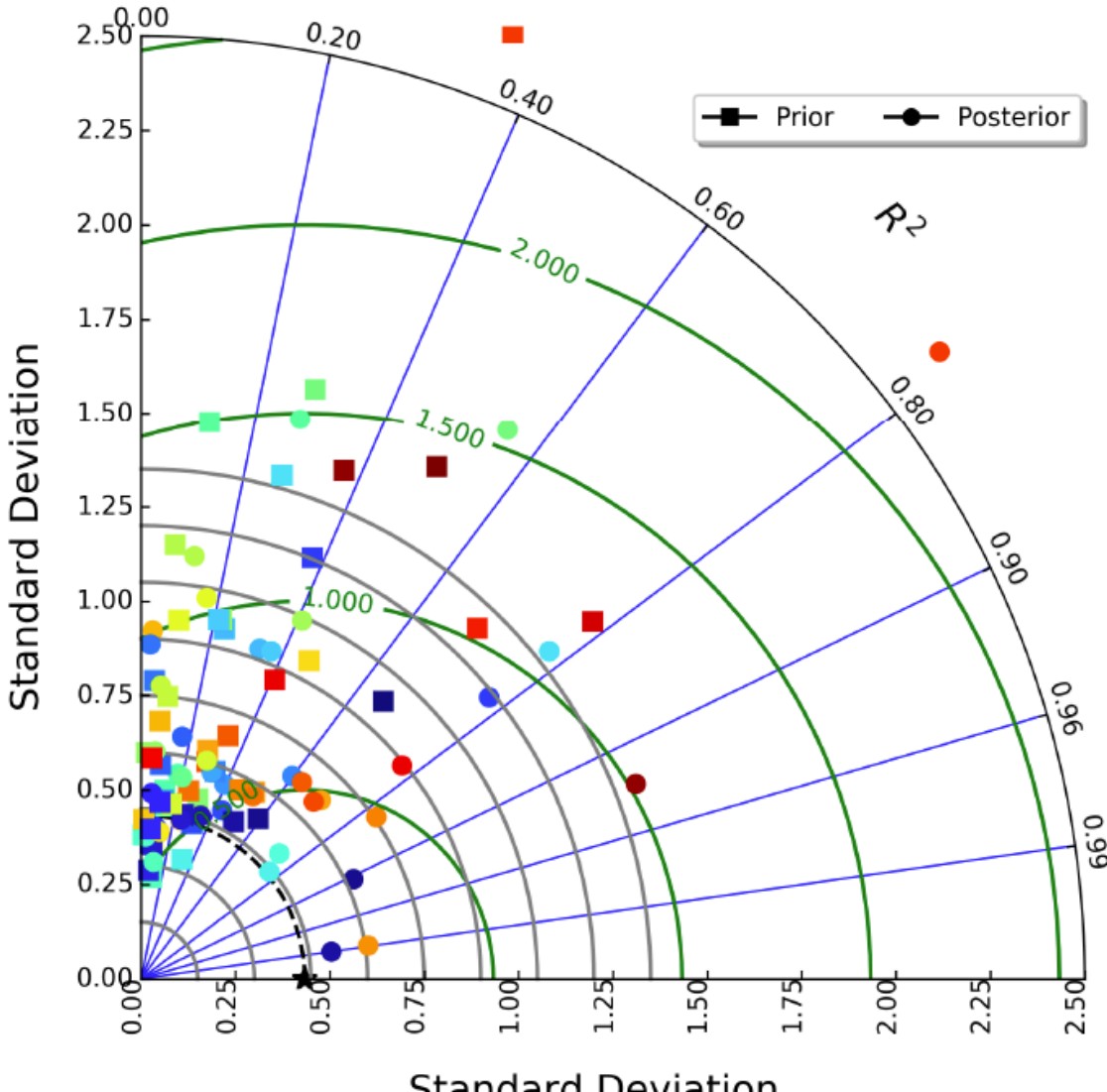

**Figure 5.** Modelled concentrations of ammonia with prior and posterior emissions against ground-based observations from 53 EMEP stations for 2013–2020 presented in a Taylor diagram. The diagram shows the Pearson's correlation coefficient (gauging similarity in pattern between the modelled and observed concentrations) that is related to the azimuthal angle (blue contours); the standard deviation of modelled concentrations of ammonia is proportional to the radial distance from the origin (black contours) and the centered normalized RMSE of modelled concentrations is proportional to the distance from the reference standard deviation (green

contours).

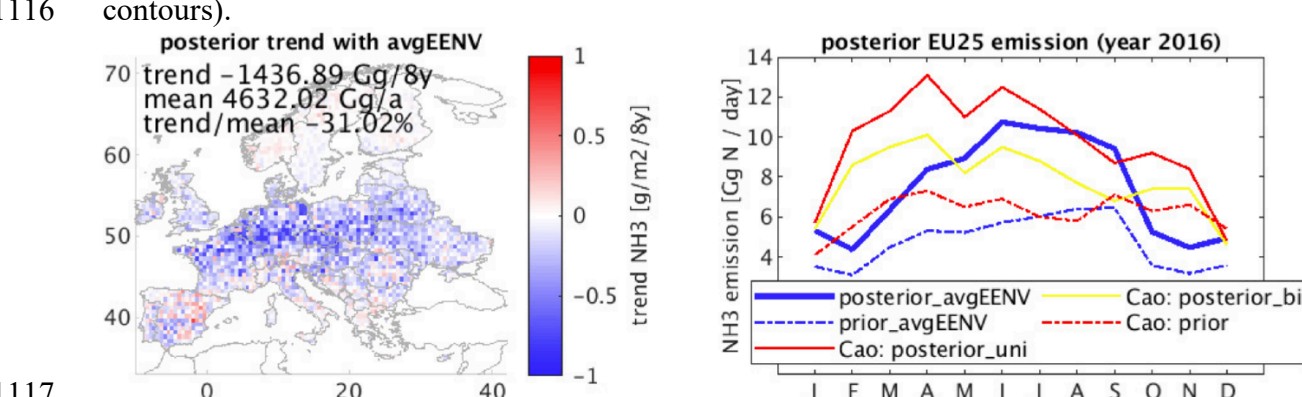


**Figure 6.** Left: spatial distribution of ammonia emission trends computed for the studied period
2013–2020 in the same way as in (Luo et al., 2022), where also trend, mean, and trend/mean
are defined/computed in the same way. Right: comparison of ammonia emissions from the
EU25 countries for the year 2016 from our posterior calculations (posterior_avgEENV, blue)
and results from Cao et al. (2022) (posterior_uni in red and posterior_bi, in yellow).