# Peer review of "Ondřej Tichý1, Sabine Eckhardt2, Yves Balkanski3, Didier Hauglustaine3, Nikolaos Evangeliou2,\"

_EGUsphere, 2023_

## Referee Comment (RC1)

**Review of "Decreasing trends of ammonia 1 emissions over Europe seen from remote sensing and inverse modelling"**

**Summary**

This paper adds to the growing body of literature on monitoring ammonia from space. The analysis uses the CFPR NH3 product derived from CrIS radiances in an inversion process that uses an LPDM within a Bayesian approach to derive monthly emissions in Europe during the 2013-2020 period. The authors show seasonal variability and calculate trends in emissions over the entire continent (excluding Russia and Turkey) and regionally, and demonstrate that overall there has been a marked decrease in emissions, attributed mainly to due to control strategies adapted by the European Union. This is an important result for policy makers to use in justifying these often unpopular controls. The use of an LPDM to tackle the problem of estimating NH3 emissions from satellite data is, to my knowledge, the first time this approach has been applied to ammonia from space.

The paper is well organized and well written. The quality of the graphics is quite high. It needs only minor revisions to be accepted for publication.

**Technical issues**

Figure 2: the posterior emissions in the upper two panels are in units ng/m2/s while they are in Gg in the two lower panels. This implies integration as well as averaging; please describe how these values were obtained in the text.

Line 292: the sentence starting with "It should be noted is not correct". If $NO_x$ and $SO_2$ decreased during the pandemic, more NH3 would remain in the atmosphere, since the there would be less sulphuric and nitric acid for it to react with. $NH_3$ emissions may well decreased in urban areas (see Cao et al., 2022 (https://pubs.acs.org/action/showCitFormats?doi=10.1021/acs.estlett.1c00730&ref=pdf) but not because $SO_2$ and $NO_X$ decreased.

Line 344: Emission may appear to increase in Scandinavia in winter because emission of OH and O3 concentrations are much lower, so the rate of converting SO2 to sulfate is much slower, less sulfate is made and thus more NH3 stays in the gas form.

Line 346: this paragraph is a bit confusing. Does the standard deviation come out of the least squares solution for Equation 4?

**Minor edits**

Line 130: 10000 retrievals per day per level seems reasonable for this product but what does the number 2920 indicate?

Line 135: … to 2000 per day per level for 6 vertical levels.

Line 177: … and one hour temporal resolution.

Line 185: the more common usage is **difference** rather than **distance**.

Line 186: between the CrIS vertical profile observations, denoted as $v^{sat}$, and the simulated retrieved profiles, $v^{ret}$. The latter are obtained by applying an instrument …….

Line 199: within a circle around each grid cell for computational efficiency. We chose circles with a radius of approximately 445 km, which is shown ….

Line 212: What are the dimensions of the matrices in this equation?

Line 218: don't the authors mean into Equation 2 and then into Equation 1?

Line 226: Does NaN here indicate missing? Please clarify.

Line 236: Does the avgEENV prior vary by year? Or is the inversion done separately each year for computational reasons?

Figure 1: temporal variability.

Line 331: Rewrite sentence starting with "Especially" as :
The NE emissions dominate the a priori emissions that were used here (avgEENV), because their winter peak in the north is extreme (emissions decline from 35 Gg in winter to 12 Gg in summer). Therefore, due to the strong prior that we use in Northern Europe there is a strong dependence of the posterior seasonality of ammonia on the prior in this region.

Line 358: The current figure 4 should come after the current figures 5 and 6.

Line 411: cannot be resolved at the spatiotemporal resolution of CTM and FLEXPART.

Line 447: A uni-directional dry deposition scheme ignores the impacts of …

Line 475: Rewrite sentence starting at: "Here we examine" as
Here we used satellite observations from CrIS and a novel inverse modelling algorithm to study the spatial variability and seasonality of NH3 emissions over Europe. We then evaluated the overall impact of such strategies on the emissions of ammonia for the period 2013–2020.

Line 481: industrial activities

Line 479: The highest emissions overall …

---

## Author Response (AR1)

**RC1**

**Review of "Decreasing trends of ammonia 1 emissions over Europe seen from remote sensing and inverse modelling"**

**Summary**

This paper adds to the growing body of literature on monitoring ammonia from space. The analysis uses the CFPR NH3 product derived from CrIS radiances in an inversion process that uses an LPDM within a Bayesian approach to derive monthly emissions in Europe during the 2013-2020 period. The authors show seasonal variability and calculate trends in emissions over the entire continent (excluding Russia and Turkey) and regionally, and demonstrate that overall there has been a marked decrease in emissions, attributed mainly to due to control strategies adapted by the European Union. This is an important result for policy makers to use in justifying these often unpopular controls. The use of an LPDM to tackle the problem of estimating NH3 emissions from satellite data is, to my knowledge, the first time this approach has been applied to ammonia from space.

The paper is well organized and well written. The quality of the graphics is quite high. It needs only minor revisions to be accepted for publication.

• We appreciate reviewer's comments and his willingness to help improve our manuscript. Below, we have done a big effort to follow his comments and answer his arguments.

**Technical issues**

Figure 2: the posterior emissions in the upper two panels are in units ng/m2/s while they are in Gg in the two lower panels. This implies integration as well as averaging; please describe how these values were obtained in the text.

• We extend Section 3.1 to describe how the totals (in Gg) are calculated (Please see TrackChanges in L.195-297).

Line 292: the sentence starting with "It should be noted is not correct". If  $NO_x$  and  $SO_2$  decreased during the pandemic, more NH3 would remain in the atmosphere, since the there would be less sulphuric and nitric acid for it to react with.  $NH_3$  emissions may well decreased in urban areas (see Cao et al., 2022

(https://pubs.acs.org/action/showCitFormats?doi=10.1021/acs.estlett.1c00730&ref=p df) but not because  $SO_2$  and  $NO_X$  decreased.

• We completely agree with the reviewer here. In fact, we have a manuscript under review saying the same thing. What we mean here is that the method we use to calculate posterior creates a false decrease in the emissions (as described in our manuscript). We rephrased (Track Changes L.334-335).

Line 344: Emission may appear to increase in Scandinavia in winter because emission of OH and O3 concentrations are much lower, so the rate of converting SO2 to sulfate is much slower, less sulfate is made and thus more NH3 stays in the gas form.

• We appreciate for this comment. We have put this explanation when discussing the seasonal variation of the emissions in N. Europe (Track Changes L.370-372)

Line 346: this paragraph is a bit confusing. Does the standard deviation come out of the least squares solution for Equation 4?

- Thank you for this question, we have reformulated this paragraph to clarify the uncertainty calculation. It is exactly as reviewer says, the natural output of the Bayesian solution (in general) is the whole posterior distribution of estimated parameters. Hence, the uncertainty calculated in Section 3.3 is the total standard deviation of the (multivariate) variable following a Gaussian distribution (Track Changes L.223-247).
- We also added the second type of uncertainty calculated from the ensemble of the used prior emissions for the inversion (Track Changes L.424).

**Minor edits**

Line 130: 10000 retrievals per day per level seems reasonable for this product but what does the number 2920 indicate?

• A typo error was corrected here (Track Changes L.132).

Line 135: ... to 2000 per day per level for 6 vertical levels.

• Corrected (please see Track Changes L.135).

Line 177: ... and one hour temporal resolution.

• Corrected (Track Changes L.180).

Line 185: the more common usage is **difference** rather than **distance**.

• Thank you, we corrected this (Track Changes L.189).

Line 186: between the CrIS vertical profile observations, denoted as  $v^{\text{sat}}$ , and the simulated retrieved profiles,  $v^{\text{ret}}$ . The latter are obtained by applying an instrument ......

• Thank you, we reformulated the sentence (see Track Changes L.191).

Line 199: within a circle around each grid cell for computational efficiency. We chose circles with a radius of approximately 445 km, which is shown ....

• Thank you, we reformulated the sentence (see Track Changes L.204).

Line 212: What are the dimensions of the matrices in this equation?

• We have clarified this in Section 2.4. The dimension for one year data batch is 193 (elements in circle around grid cell) times 6 (vertical profiles) times 12 (months), hence 13896 (see Track Changes L.229-231).

Line 218: don't the authors mean into Equation 2 and then into Equation 1?

• It is, indeed, into Eq. 2 and then Eq. 1. We reformulate the sentence to omit misunderstanding (see Track Changes L.254).

Line 226: Does NaN here indicate missing? Please clarify.

• Indeed, we have replaced "NaN" by "missing" (see Track Changes L.286).

Line 236: Does the avgEENV prior vary by year? Or is the inversion done separately each year for computational reasons?

• Both notes are correct. The avgEENV prior vary by year because some priors vary by year. Also, the inversion is done separately for each year which is now commented more clearly in Section 2.4 (see Track Changes L.212).

Figure 1: temporal variability.

• Corrected (see Track Changes L.1018).

Line 331: Rewrite sentence starting with "Especially" as :

The NE emissions dominate the a priori emissions that were used here (avgEENV), because their winter peak in the north is extreme (emissions decline from 35 Gg in winter to 12 Gg in summer). Therefore, due to the strong prior that we use in Northern Europe there is a strong dependence of the posterior seasonality of ammonia on the prior in this region.

• We rewrote the sentence (see Track Changes L.404-406).

Line 358: The current figure 4 should come after the current figures 5 and 6.

• Corrected everywhere in the manuscript (see Track Changes).

Line 411: cannot be resolved at the spatiotemporal resolution of CTM and FLEXPART.

• Corrected (see Track Changes L.556).

Line 447: A uni-directional dry deposition scheme ignores the impacts of ...

• Part of this sentence removed as reviewer suggested (see Track Changes L.597).

Line 475: Rewrite sentence starting at: "Here we examine" as

Here we used satellite observations from CrIS and a novel inverse modelling algorithm to study the spatial variability and seasonality of NH3 emissions over Europe. We then

evaluated the overall impact of such strategies on the emissions of ammonia for the period 2013–2020.

• Sentence modified as suggested (see Track Changes L.628-631).

Line 481: industrial activities

• Corrected (see Track Changes L.634).

Line 479: The highest emissions overall ...

• We were not sure if we understood what is meant to be written here and modified the sentence as "The highest emissions over the 2013–2020..." (see Track Changes L.632).

**RC2**

In their paper Tichy et al present the emissions of ammonia derived from CrIS satellite observations, and present the trend in these emissions over the period 2013-2020. To my judgement major revisions are needed before the paper can be published, as detailed in the comments below.

• We appreciate reviewer's assistance to improve our manuscript. We have his comments in an effort to optimize our paper.

**General comments:**

In general the description of the method, inputs, filtering, error modeling are incomplete in the paper, and make it impossible to judge the quality of the results, in particular the reported trends, but also the absolute value of the emissions.

- We have modified the methodology description, adding more information. However, please note that we have intentionally omitted many details to avoid repetition. Our first paper in which we used the same set-up stands as a preprint. Although we do not have a revised or a final version yet, the preprint is cited whenever needed within the manuscript (Tichý et al., 2022). Please see section 2.4.
- As regards to the uncertainty, we now provide more details about how it was calculated, plus that we add another approach that is based on the use of an ensemble of different priors to calculate how sensitive our posterior emissions are with respect to the use of prior (Please see section 3.3 of the Track Change Manuscript).

Tichý, O., Otervik, M. S., Eckhardt, S., Balkanski, Y., Hauglustaine, D. and Evangeliou, N.: NH3 levels over Europe during COVID-19 were modulated by changes in atmospheric chemistry, npj Clim. Atmos. Sci., in review, 1–13, doi:10.21203/rs.3.rs-1930069/v1, 2022.

Trends in ammonia are presented without discussing other trace gases, in particular NOx and SO2, which have a significant trend over the past decade and influence ammonia concentrations. No NOx/SO2 trend results are shown in the paper, and the authors do not provide evidence that the model used (LMDZ-OR-INCA) provide a realistic description of trends and interaction with other chemicals and aerosols.

- We are not sure how to act on this comment. LMDz-OR-INCA model has been used the last 20 years. Its chemistry was implemented by our coauthor and we cite all the details of it in Hauglustaine et al. (2004 and 2012). We do not intend to evaluate the chemical scheme of LMDz-OR-INCA nor repeat technical details here. Both have been done long before and are presented in the references we cite. However, if the Reviewer and Editor insist, we could add a few repetitive sections.
- Also, we evaluate the trends in the emissions and not the trends of atmospheric ammonia in general. Although they are of course linked, NOx/SO2 have been taken by published state-of-the-art inventories (ECLIPSE in the present). As it is explained in our preprint (Tichy et al., 2022) and here too, the chemical loss of NH3 due to chemical reactions with sulfate and nitrate (products of NOx/SO2) is introduced in flexpart with the e-folding lifetime. In other words, we calculate the

lifetime of NH3 in the chemical model first, which then import as a loss parameter in flexpart. All the calculations of the modelled lifetime are presented in our previous work (Evangeliou et al., 2021).

Hauglustaine, D. A., Hourdin, F., Jourdain, L., Filiberti, M.-A., Walters, S., Lamarque, J.-F. and Holland, E. A.: Interactive chemistry in the Laboratoire de Meteorologie Dynamique general circulation model: Description and background tropospheric chemistry evaluation, J. Geophys. Res., 109(D04314), doi:10.1029/2003JD003957, 2004. Hauglustaine, D. A., Balkanski, Y. and Schulz, M.: A global model simulation of present and future nitrate aerosols and their direct radiative forcing of climate, Atmos. Chem. Phys., 14(20), 11031–11063, doi:10.5194/acp-14-11031-2014, 2014.

Tichý, O., Otervik, M. S., Eckhardt, S., Balkanski, Y., Hauglustaine, D. and Evangeliou, N.: NH3 levels over Europe during COVID-19 were modulated by changes in atmospheric chemistry, npj Clim. Atmos. Sci., in review, 1–13, doi:10.21203/rs.3.rs-1930069/v1, 2022.

Evangeliou, N., Balkanski, Y., Eckhardt, S., Cozic, A., Van Damme, M., Coheur, P.-F., Clarisse, L., Shephard, M., Cady-Pereira, K. and Hauglustaine, D.: 10–Year Satellite– Constrained Fluxes of Ammonia Improve Performance of Chemistry Transport Models, Atmos. Chem. Phys., 21, 4431–4451, doi:10.5194/acp-21-4431-2021, 2021.

How much does the a-priori emission influence the results? The method description in section 2.4 does not provide the information to judge the influence of the prior compared to the impact of the satellite measurements.

• We extended section 2.4 and completely rewrote section 3.3 to address the question of uncertainty more clearly (Please see the provided manuscript with Track Changes). In the revised version of the manuscript, we study two types of uncertainty, the inversion model uncertainty and the uncertainty arising from prior emissions. The model uncertainty is a natural result from the used Bayesian approach where full Gaussian posterior distribution is available and uncertainty in the form of variance/standard deviation can be easily calculated. The prior emissions uncertainty is calculated using ensemble of the used prior emissions.

The assumptions/modeling of errors of the satellite data (including filtering), in the method (model uncertainty: chemistry, transport) and a-priori emissions are not described. Section 3.3 discusses uncertainties but is very high level and does not provide the details needed to understand the results and related error bars.

• Thank you for pointing out the influence of prior emissions. We rewrote Section 3.3 completely to clearly describe the calculation of uncertainty from the posterior estimates based on property of Gaussian distribution which is the form of our posterior. Moreover, we added the second type of uncertainty calculated from the ensemble of the used prior emissions for the inversion. The abstract and conclusion parts are modified accordingly.

Satellite observations are available once per day, but I assume that the emissions are reported as diurnal mean. What uncertainty does the unknown diurnal cycle introduce?

• We are not sure how to quantify this type of uncertainty and we do not have any indication whether it is significant of not. A diurnal cycle is neither assumed in the Chemistry Transport Model, nor exists in the satellite observations from CrIS,

as pointed by the reviewer. If the Editor or Reviewer think it is necessary to quantify, we would be happy to do it, and await specific instructions.

**Detailed comments:**

The abstract is long and reads like an introduction, especially the first part. I would propose to shorten it and focus on the actual findings in the paper and new results.

• We agree and have shortened the abstract to 306 words (Please see Track Changes).

The paper has a good introduction with a balanced set of relevant papers.

• We appreciate again for the time the reviewer took to read and try to improve our manuscript.

The paper uses several units for the emissions (per second, per day, per month, per year). This makes it hard to compare the plots. I would suggest to restrict this to one or two choices.

• We have tried to do this; however, there are cases where we cannot have full control of the units. For instance, when comparing our findings with results from the literature, we have to follow the units presented in the literature.

l 40: "Our results are associated with relatively low uncertainties reaching a maximum of 42%" Which result is this? Is it the trend over a region?

• This sentence is now reformulated to address both Gaussian model uncertainty and prior emissions ensemble uncertainty, respectively. Note also that we have updated the Gaussian posterior model uncertainty with higher precision reaching relative uncertainty value.

l 47: "constitute a robust basis for European NH3 estimates". What does "robust basis" refer to. Do the authors claim that the monitoring of pollution levels as set by the regulations can be performed based on satellite observations and inverse modeling (only)?

• We have modified the abstract completely so that it is much shorter now. We explicitly write in L38-40 that "These results indicate that satellite measurements combined with inverse algorithms constitute a robust tool for emission estimates and can infer the evolution of ammonia emissions over large timescales".

l 47: "de facto". Does this mean that the evolution is based on measurements?

• We have removed this confusing expression (see Track Changes L38-40).

187: "Greenhouse Gases Observing Satellite". Please add acronym "GOSAT".

• The acronym has been added (Track Changes L.147).

l 95: "using alternation between CrIS ammonia retrievals performed with the logarithm of concentrations and linearized retrievals." Please explain more clearly: what does the "alternation" between log and linear retrievals mean?

• We completely reformulated this sentence to be more precise in referring to the method by Sitwell et al. (2022) (Track Changes L.155-157).

l 97: "use direct comparisons between the CrIS observations and model retrievals". What are "model retrievals"?

• The sentence has been corrected as to "comparison between CrIS ammonia retrievals and model profiles" (Track Changes L.158).

l 118: "total column random measurement error is estimated in the 10–15% range, with total random errors estimates of  $\sim$ 30%". What is the difference between a "total column random" and a "total random" error? Systematic errors are even more important.

 As Shephard et al. (2020) reported "For the total column amounts, the measurement errors are typically in the 10% to 15% range, whereas the total errors are ~ 30 %". We now correct for this typo in L.185 (see Track Changes).

Shephard, M. W., Dammers, E., E. Cady-Pereira, K., K. Kharol, S., Thompson, J., Gainariu-Matz, Y., Zhang, J., A. McLinden, C., Kovachik, A., Moran, M., Bittman, S., E. Sioris, C., Griffin, D., J. Alvarado, M., Lonsdale, C., Savic-Jovcic, V. and Zheng, Q.: Ammonia measurements from space with the Cross-track Infrared Sounder: Characteristics and applications, Atmos. Chem. Phys., 20(4), 2277–2302, doi:10.5194/acp-20-2277-2020, 2020.

l 121: "due to the limited vertical resolution". Please mention a typical degrees of freedom of signal for CrIS NH3.

• Missing information were added in L.188 (see Track Changes).

l 130: "Daily CrIS ammonia (version 1.6.3) was interpolated onto a  $0.5^{\circ} \times 0.5^{\circ}$  grid ". How is this done? How are measurements and kernels, defined in log space, averaged? It seems to me that this needs to be done with great care, so more details are required to convince the reader of the correctness of the approach.

- The averaging has been performed under the guidance of the CrIS developers. While we initially thought to use classic oversampling methods, the resolution we wanted to achieve is very coarse, while oversampling is more efficient in urban scales (Zhu et al., 2014). Also, we had tested previously inverse distance weighting interpolation (IDW) in Evangeliou et al. (2021) finding that it creates overestimated gridded NH3 columns.
- Therefore, we decided to use classic linear interpolation and validate our results against ground-based observations from EMEP, which are openly available from <a href="https://ebas.nilu.no">https://ebas.nilu.no</a>. Some examples are shown in Fig. R1-R4.

Zhu, L., Jacob, D.J, Mickley, L.J., Marais, E.A., Cohan, D.S., Yoshida, Y., Duncan, B.N., Abad, G.G., and Chance, K.V.: Anthropogenic emissions of highly reactive volatile organic compounds in eastern Texas inferred from oversampling of satellite (OMI) measurements of HCHO columns. Environ. Res. Lett. 9, 114004, 2014. Evangeliou, N., Balkanski, Y., Eckhardt, S., Cozic, A., Van Damme, M., Coheur, P.-F., Clarisse, L., Shephard, M., Cady-Pereira, K. and Hauglustaine, D.: 10–Year Satellite– Constrained Fluxes of Ammonia Improve Performance of Chemistry Transport Models, Atmos. Chem. Phys., 21, 4431–4451, doi:10.5194/acp-21-4431-2021, 2021.

Fig. R 2

---

## Author Response (AR2)

Anonymous during peer-review: Yes No

Anonymous in acknowledgements of published article: Yes No

**Checklist for reviewers**

**1) Scientific significance Outstanding Excellent Good Fair Low Does the manuscript represent a substantial contribution to scientific progress within the scope of this journal (substantial new concepts, ideas, methods, or data)? 2) Scientific quality Outstanding Excellent Good Fair Low Are the scientific approach and applied methods valid? Are the results discussed in an appropriate and balanced way (consideration of related work, including appropriate references)? 3) Presentation quality Outstanding Excellent Good Fair Low Are the scientific results and conclusions presented in a clear, concise, and well structured way (number and quality of figures/tables, appropriate use of English language)?**

**For final publication, the manuscript should be**

**accepted as is**

accepted subject to **technical corrections** accepted subject to **minor revisions** reconsidered after **major revisions rejected**

**Were a revised manuscript to be sent for another round of reviews:**

I would be willing to review the revised manuscript.

I would not be willing to review the revised manuscript.

**Suggestions for revision or reasons for rejection**

(visible to the public if the article is accepted and published) Paper is acceptable as is for publication.

Response: We appreciate reviewer's help to improve our manuscript.

Anonymous during peer-review: Yes No

Anonymous in acknowledgements of published article: Yes No

**Checklist for reviewers**

| 1) Scientific significance
Does the manuscript represent a substantial
contribution to scientific progress within the
scope of this journal (substantial new
concepts, ideas, methods, or data)?                | Outstanding Excellent Good Fair Low |
|------------------------------------------------------------------------------------------------------------------------------------------------------------------------------------------------------------------------------------|-------------------------------------|
| 2) Scientific quality
Are the scientific approach and applied
methods valid? Are the results discussed in an
appropriate and balanced way (consideration
of related work, including appropriate
references)? | Outstanding Excellent Good Fair Low |
| 3) Presentation quality
Are the scientific results and conclusions
presented in a clear, concise, and well
structured way (number and quality of
figures/tables, appropriate use of English
language)?       | Outstanding Excellent Good Fair Low |
| For final publication, the manuscript should be                                                                                                                                                                                    |                                     |
| accepted as is                                                                                                                                                                                                                     |                                     |

accepted subject to technical corrections

**accepted subject to minor revisions**

reconsidered after major revisions

rejected

**Were a revised manuscript to be sent for another round of reviews: I would be willing to review the revised manuscript.**

I would not be willing to review the revised manuscript.

**Suggestions for revision or reasons for rejection**

First of all I thank the authors for their detailed replies to my comments and changes made to the manuscript. These have, to my opinion, improved the manuscript considerably, in particular the rewritten sections 2.4, 3.3 and in particular section 4.2 address several of my comments. The addition of the uncertainty estimate from the emission ensemble spread is also of interest. But I still have a couple of remaining and unanswered comments which I would suggest the authors take into account before the paper is published.

**Response:** We have tried to answer all comments posed from the reviewer step-by-step in the manuscript with Track Changes enabled.

The Tichy 2022 preprint on the COVID-19 impact on ammonia nicely shows the importance of SO2 and NO2 in determining the lifetime of ammonia. The COVID-19 reductions in NOx and SO2 emissions have a major impact on the lifetime, as mentioned in the last lines of section 3.1 (which is neglected in the current paper). And lifetime has a major impact on the emission estimate. For trend estimates it is clear that a wrong representation of trends in NOx and SO2 will lead to systematic biases in the estimated trends. So it is very important to demonstrate that the modelling system correctly represents the trends in NO2 and SO2. As mentioned by the authors use is made of stateof-the-art inventories, which provides some trust in the trends presented. But the realism of the lifetime estimate could be demonstrated by comparing modelled NO2/SO2 time series (trends) with surface observations of these two species, and by presenting this as for instance an extra figure in the supplement. In the paper NOx and SO2 are only mentioned in the disclaimer on COVID-19, but are not discussed at all in the results/discussion parts of the paper. My major comment was about this lack of discussion in the paper about this key factor in the trend estimate. Please add a discussion and possibly observational evidence that the NOx/SO2 levels and trends are reasonalbly well modelled in the system.

**Response:** We understand the concerns reported by the reviewer and we have now added an extended validation of NO2 and SO2 as requested. We have added 2 example plots validating ground measurements of SO2 and NO2 (respectively) against modelled concentrations in random stations and scatterplots of the full dataset used for the validation (Suppl. Figure S 8 and S 9). We also discuss the model validation with regard to NO2/SO2 concentrations in the paragraphs starting at P.15-L.479.

My major comment "The assumptions/modeling of errors of the satellite data (including filtering), in the method (model uncertainty: chemistry, transport) and a-priori emissions are not described" has not been fully addressed.

**Response:** We have tried to address the comments from the reviewer in the points below.

However, please note that uncertainty of the prior emissions has not been calculated in detail. Klimont et al. (2017) (http://www.atmos-chem-phys-discuss.net/acp-2016-880/) reported "We have not performed a formal uncertainty analysis for emission estimates in this study, but results of analysis from other studies are helpful and indicative of the expected un- certainties for various species and regions. For example, the global BC and OC inventory developed by Bond et al. (2004) included an uncertainty analysis of total emissions providing regional "low-high" estimates for 1996. For BC emissions from

**anthropogenic sources, the range was 3.1-10 Tg yr-1 (-30 to +120 %) and for OC 5.1-14 Tg yr-1 (-40 to +130 %)."**

1. Satellite data: The satellite data is "interpolated" to a 0.5 by 0.5 degree grid, reducing the number of observations. Details on how this is done are missing. A linear interpolation is mentioned. But are the kernels and covariance matrices interpolated in the same way? Please explain how this is done. Is the error (covariance) of a weighted mean concentration of various observations equal to the weighted mean of the covariances? In other words: are nearby CrIS observations correlated or uncorrelated? Is the full error covariance of CrIS taken into account (errors are strongly correlated)? Please specify how the satellite observational operator has been generated "in a robust manner."

**Response:** We have changed the term "interpolation" to "gridding", because it represents better what we did and we further explain in detail the procedure (Track Changes P.5-L.133-140). Furthermore, we give 8 supplementary figures (Supplementary Figure S 1), corresponding to 1st September of each of the 2013-2020 years of our study period, showing the quality of the gridding with respect to raw kernel data, gridded on 0.5 degrees over Europe, and the calculated standard deviation of the values falling within each 0.5 degree grid-cell. The calculated low standard deviation as compared to the gridded AK values show that the raw kernel values were very similar in each grid-cell causing small bias.

2. Modelling uncertainties: Are the grid-cell specific SRMs assumed to be free of error? Section 2.4 provides technical details on the implementation but does not discuss input assumptions for the model-related uncertainties. R in eq.5 is modelled with an omega factor and I\_p. What is this I\_p? Is it a diagonal unity matrix (are correlations between the vertical levels in CrIS accounted for)?

**Response:** I\_p is the identity matrix with ones on its diagonal and zeros otherwise, while factor omega is estimated by the model within the calculation procedure. This is now clarified in the revised version of the manuscript (please see manuscript with Track Changes at P.8 – L.220-236).

3. Emissions: A cost function (equation 3) normally contains a term reflecting the apriori uncertainty (of the emissions). This seems to be missing here. Are emission uncertainties taken into account in the optimal estimation?

**Response:** The optimization formulation in Eq. (3) and its restriction to the surroundings in Eq. (4) is formulated without uncertainty terms. However, a probabilistic model is employed exactly for the introduction of uncertainties into the inference. Here, we refer to Tichý et al. (2020) paper, where equivalence between classical optimization formulation and its probabilistic counterpart is shown, with the benefit of a probabilistic model, which can adaptively estimate other parameters of the model such as the covariance matrices, namely omega, L and V. Note also that the structure of the covariance matrix, e.g., in Eq. (7) is our prior assumption, however, its posterior structure can differ from the prior structure significantly. Typically, the posterior structure is a full matrix, see Appendix B in Tichý et al. (2020) for more details on posterior parameters calculation.

In the reply the authors mention "A diurnal cycle is neither assumed in the Chemistry Transport Model, nor exists in the satellite observations from CrIS". Please mention this

point explicitly in the discussion section of the paper. The monthly-mean emissions probably need to be interpreted as montly-mean emissions at the satellite overpass time, which will differ from the 24h daily mean emission.

**Response:** We have added this statement in Track Changes P.9-L.274-276. This is a common problem in many similar studies that involve satellite observations. We hope this specific comment will not cause further confusion to readers.

I did not understand equation 9. Please define the quantities in this equation (sigma\_location). Does the square root cover the whole expression? I would expect a 1/n normalisation before the summation.

Response: We thank the reviewer for this question/comment. We reformulated equation (9) by replacing the  $\Sigma_{\text{location}}$  to the  $\sigma_{\text{location}}$  to avoid confusion with the sum symbol used in the right side of the equation (Track Changes P.13-L388). Please note also that there is no normalization before the summation. This arises as a property of the sum of Gaussian distributions. For Gaussian distributed independent variables X\_1, X\_2, ..., X\_n with means mu\_1, mu\_2, ..., mu\_n and variances sigma\_1^2, sigma\_2^2, ..., sigma\_n^2, their sum follows the normal distribution: sum\_{i=1}^n X\_i = N(sum\_{i=1}^n n mu\_i, sum\_{i=1}^n sigma\_i^2) When reporting the standard deviation rather than variance, the standard deviation is square root of the variance, hence sqrt( sum\_{i=1}^n sigma\_i^2 ), which is the core of

equation (9).

Sec 2.1, line 157 (v2 manuscript) mentions "The individual profile retrieval levels show an estimated random measurement error of 10-30 %, with total random errors estimates increasing to 60 to 100%" What is the difference between " estimated random and total random errors?

**Response:** All the different metrics that are mentioned in this section were requested to be written here by the researchers involved in CrIS product/retrieval developments. Each of them is explained in Shephard et al. (2015), Shephard and Cady-Pereira (2015), Shephard et al. (2020) [all references can be found in the reference list of the manuscript]. All the metrics are explained there in detail and are also given in the product as shown in their metadata below.

If the reviewer and/or editor believe they are disturbing and/or not useful, we could remove some of the statistics:

```
netcdf Combined_NH3_p165_0_p180_0_n050_0_n045_0_20200101 {
dimensions:
    Observations = 1043 ;
    Layers = 15 ;
    RVMRLen = 5 ;
    nerr = 2 ;
variables:
    double xretv_meas_error(Observations, Layers, nerr) ;
        xretv_meas_error:units = "ppmv" ;
        xretv_meas_error:long_name = "Retrieved Species Measurement Error:
(Minus,Plus)" ;
...
double rvmr_error(Observations, RVMRLen) ;
```

```
rvmr_error:units = "ppmv" ;
```

| rvmr_error:long_name = "RVMR Error " ;                                |
|-----------------------------------------------------------------------|
|                                                                       |
| double xretv_total_error(Observations, Layers, nerr) ;                |
| xretv_total_error:units = "ppmv" ;                                    |
| xretv_total_error:long_name = "Retrieved Species Total Error:         |
| (Minus,Plus)";                                                        |
| double tot_col_meas_error(Observations);                              |
| tot_col_meas_error:units = "molec/cm2" ;                              |
| tot_col_meas_error:long_name = "Retrieved Species Total Column        |
| Measurement Error" ;                                                  |
| double tot_col_total_error(Observations);                             |
| tot col total error:units = "molec/cm2";                              |
| tot col total error:long name = "Retrieved Species Total Column Total |
| Error";                                                               |
| ·····                                                                 |
| double total_covariance_error(Observations, Layers, Layers) ;         |
| total covariance error:units = "ln(vmr)^2":                           |

total\_covariance\_error:long\_name = "The total error covariance matrix is the sum of smoothing and measurement error (systmatic not included at this time). For atmospheric temperature, it represents the covariance of the error of temperature. For Atmospheric Species, it is the covariance of the error of ln(vmr)";

...

double noise\_error\_covariance(Observations, Layers, Layers) ; noise\_error\_covariance:units = "ln(vmr)^2" ;

noise\_error\_covariance:long\_name = "The Measurement error covariance matrix from the radiances. Presently it is also used as a lower limit on the Observation error covariance matrix (Measurement + systematic + cross-state errors) as systematic and cross-state are not explicitly derived at this time. The utility of the observation error is for comparisons with other measurements and for assimilation. The smoothing error is accounted for when one applies the averaging kernel, so the observation error accounts for everything else." ;

l 387: "2.410The calculated posterior uncertainty for our spatial domain and studied period (2013–2020) is shown in Figure 4Figure 4 (right)." Please correct. **Response:** It has been corrected now (Track Changes P.14-L.431).

The new section 4.2 is quite long and may be condensed (to 1 page if possible) to become more in balance with the other sections. A few suggestions: The first paragraph is long, with some repetition, and may be reduced to a few lines. Fig. S8 could be removed and the second paragraph could be summarised in one line. **Response:** We have tried to remove repetition from different parts of section 4.2 (see Track Changes P.18). Since this section justifies why this method is suitable for NH3 calculations and answers why a more classic method was not used, we would prefer to keep Supplementary Figures as in the previous version, because they are explanatory and – besides – they do not occupy additional space in the main manuscript.

For some stations the differences between the forward/backward calculations and LMDz are quite large, e.g. at DE0002R. Does this have implications for the emission

estimates? Could you provide a (rough) estimate of how much these modelling differences influence the uncertainty of the a-posteriori emissions?

**Response:** We have added a full explanation why this happens in Lagrangian simulations (see Track Changes P.20 – L.665-673). The main message, from the explanation given in the manuscript, is that backward simulations are always more accurate than forward ones, mainly due to the larger number of particles given per release (two more reasons are also described in P.20).

For example, in forward simulations, where there are hundred thousand of releases, the number of particles is weighted with the mass (the larger the mass, the larger the particle number), until an upper limit of 400 million particles is reached (larger number causes memory allocation problems). This set-up results in particle numbers of up to 1-2 thousand particles per release, at maximum. Backward simulations use 50 thousand particles per release (25-50 times more particles), due to the limited release points (one for each receptor), as usually releases occur at specific receptors (in the present case, at stations where we had observations).

Hence, no additional modelling uncertainty is expected in the posterior emissions of NH3.